# OBCache: Optimal Brain KV Cache Pruning for Efficient Long-Context LLM Inference

## Abstract

Large language models (LLMs) with extended context windows enable powerful applications but impose significant memory overhead, as caching all key–value (KV) states grows linearly with sequence length and batch size. Existing cache eviction methods address this by exploiting attention sparsity, yet they typically rank tokens heuristically using accumulated attention weights without considering their true impact on attention outputs. We propose Optimal Brain Cache (OBCache), a principled framework that formulates cache eviction as a layer-wise structured pruning problem. Building on Optimal Brain Damage (OBD) theory, OBCache quantifies token saliency by measuring the perturbation on attention outputs induced by pruning tokens, with closed-form scores derived for isolated keys, isolated values, and joint key–value pairs. Our scores account not only for attention weights but also for information from value states and attention outputs, thereby enhancing existing eviction strategies with output-aware signals. Experiments on LLaMA and Qwen models show that replacing the heuristic scores in existing works, which estimate token saliency across different query positions, with OBCache's output-aware scores consistently improves long-context accuracy.

## 1 Introduction

Large language models (LLMs) (Touvron et al., 2023a;b; Bai et al., 2023a; OpenAI, 2023) have recently revolutionized a wide range of natural language processing tasks, including document summarization (Zhang et al., 2024a), question answering (Kamalloo et al., 2023), code generation (Roziere et al., 2023), and dialogue systems (Taori et al., 2023; Chiang et al., 2023). Despite their impressive capabilities, many of these applications require processing long sequences, which poses substantial challenges for efficient deployment. A major bottleneck in LLM inference stems from their autoregressive nature, which necessitates caching all key-value (KV) states across the context window. Because the cache size scales linearly with both sequence length and batch size, it leads to substantial memory and latency overheads. For instance, running a LLaMA-3.1-8B model (Grattafiori et al., 2024) with a 1M-token context window requires storing over 120GB of KV cache, which exceeds the memory capacity of most GPUs.

A promising line of research addresses this challenge via *KV cache eviction*, a training-free technique that reduces inference costs by selectively discarding unimportant KV tokens (Zhang et al., 2023; Xiao et al., 2024). These methods are motivated by the observation that only a small subset of tokens significantly influences model predictions. By evicting redundant tokens during inference, such approaches can substantially reduce memory and computational costs while incurring moderate performance degradation. Early methods such as $H_2O$ (Zhang et al., 2023) evict tokens based on accumulated attention weights, a simple yet effective heuristic for estimating token saliency. More recent techniques, including TOVA (Oren et al., 2024) and SnapKV (Li et al., 2024), refine this strategy by introducing more sophisticated attention-based scoring mechanisms to better preserve accuracy. However, these methods rely primarily on attention weights and often overlook the contribution of value states in shaping the final model outputs. Heuristically accumulating attention scores fails to fully capture the true impact of token removal on attention outputs, which directly affects the hidden states and downstream predictions. Consequently, these methods may mistakenly retain tokens with negligible influence or discard ones whose importance is not evident from attention weights alone.

To tackle these limitations, we propose Optimal Brain Cache (OBCACHE), a principled framework that formulates KV cache eviction as a layer-wise structured pruning problem. The key insight behind OBCACHE is that the actual impact of removing KV pairs, i.e., their influence on future model outputs, can be effectively approximated by analyzing local perturbations in historical attention outputs when the corresponding KV vectors are pruned. This approximation enables us to estimate the contribution of each token to the model output without requiring access to future states at inference time. Our analysis is grounded in the Optimal Brain Damage (OBD) theory (LeCun et al., 1989), originally proposed for pruning model weights. By treating cached keys and values as pruning variables, we define three types of pruning units: isolated value vectors, isolated key vectors, and joint key-value pairs at the same token position. In each case, we derive closed-form expressions for the pruning-induced output perturbation through second-order Taylor approximation. These perturbation estimates serve as token-wise saliency scores to inform cache eviction and retention decisions.

In contrast to prior approaches, OBCACHE scores incorporate not only attention weights but also value states, pre-softmax attention logits, and attention outputs. This results in output-aware saliency measures that provide richer and more accurate signals for cache eviction. Furthermore, we show that existing attention-based scoring methods emerge as special cases under our framework. Specifically, the pruning objective is simplified to preserving the attention matrix, and the pruning units are reduced to individual attention columns. As such, OBCACHE generalizes and complements existing cache eviction methods, and can be seamlessly integrated into any score-based cache eviction pipeline to improve token selection.

We empirically demonstrate the effectiveness of OBCACHE through extensive experiments on long-context benchmarks. Specifically, we incorporate the three OBCACHE saliency scores into existing cache eviction frameworks, including H2O, TOVA, and SnapKV. Across both LLaMA-3.1 and Qwen-2.5 models, we observe consistent performance improvements on a variety of long-context tasks, including Needle-in-a-Haystack passkey retrieval, long-sequence perplexity evaluation, and 16 benchmarks from LongBench (Bai et al., 2023b). Our results demonstrate that integrating OB-CACHE scores can produce more accurate estimations of token saliency, significantly enhancing long-context inference performance while maintaining low computational overhead.

In summary, our key contributions are as follows:

- We introduce OBCACHE, a principled scoring framework for KV cache eviction that directly targets eviction-induced perturbations in attention outputs. The framework provides output-aware saliency measures that can be seamlessly integrated into existing cache eviction pipelines to improve token selection and saliency estimation.

- We provide the first theoretical formulation of KV cache eviction as a structured pruning problem, based on the Optimal Brain Damage (OBD) framework. Using second-order Taylor approximations, we derive closed-form perturbation estimates for isolated keys, isolated values, and joint key–value pairs. This theoretical analysis also clarifies the limitations of prior attention-weight heuristics and shows that they can be viewed as special cases of our more general formulation.

- We conduct extensive empirical studies demonstrating that replacing the heuristic scores in existing methods ($H_2O$, TOVA, SnapKV), which estimate token saliency across different query positions, with OBCACHE's output-aware scores consistently improves performance on both LLaMA and Qwen models across diverse long-context benchmarks, including retrieval, perplexity evaluation, and LongBench tasks.

## 2 RELATED WORKS

**KV Cache Compression.** In long-context scenarios, reducing the size of the key–value (KV) cache is critical for optimizing the deployment of large language models (LLMs). To this end, *cache eviction* methods are motivated by the observation that only a sparse subset of tokens can significantly contribute to model predictions. For example, StreamingLLM (Xiao et al., 2024) identifies the *attention sink* phenomenon and retains both the initial and most recent tokens to enable infinite-context decoding. $H_2O$ (Zhang et al., 2023) proposes accumulating attention weights across all query positions to dynamically identify salient tokens. TOVA (Oren et al., 2024) simplifies this by considering only the attention distribution of the most recent query. A more refined strategy,

SnapKV (Li et al., 2024), aggregates attention scores within a small observation window and applies a pooling-based clustering mechanism, which improves performance in retrieval-centric tasks. Orthogonal to eviction-based methods, Quest (Tang et al., 2024) retains the full KV cache and restricts attention computation to the most relevant tokens. CaM (Zhang et al., 2024b) and $D_2O$ (Wan et al., 2025) aim to mitigate the information loss caused by irreversible eviction and merge evicted KV states into the retained cache. However, none of these approaches explicitly models the contribution of value states in estimating token importance, nor do they quantify the direct impact of KV removal on model outputs. In contrast, our method evaluates token saliency through output-aware perturbation analysis, providing a more principled and generalizable framework for cache eviction.

**Model Pruning.** Another direction for reducing LLM inference costs focuses on pruning model parameters. Classical pruning frameworks (LeCun et al., 1989; Hassibi & Stork, 1992) quantify the saliency of each pruning unit by estimating the perturbation it induces in a task-specific loss, often approximated to second order using a Taylor series. However, computing second-order statistics globally is still inefficient in LLM-scale models (Ma et al., 2023). To reduce complexity, recent approaches adopt local formulations that operate at the level of individual layers (Frantar & Alistarh, 2023; Sun et al., 2024), where the objective becomes minimizing the change in layer output. By varying the pruning units, such methods can structurally remove redundant components such as attention layers, heads, or hidden channels. In this work, we extend the layer-wise pruning paradigm to handle the eviction of the dynamic KV cache. By treating KV states as pruning units, we introduce a theoretically grounded framework to more accurately quantify token saliency.

## 3 OBCACHE

In this section, we present the details of OBCACHE. We begin by reviewing the workflow of key–value (KV) caching and formalizing the notation in Section 3.1. In Section 3.2, we formulate cache eviction as a layer-wise structured pruning problem, with the objective of minimizing perturbations in attention outputs. Section 3.3 details our analytical solution to this perturbation minimization problem based on second-order Taylor approximations. In Section 3.4, we show that our framework recovers existing attention-based scoring methods as special cases. We then provide a qualitative example to highlight the effectiveness of our formulation in Section 3.5. An overview of the OBCACHE scoring mechanism is shown in Figure 1.

### 3.1 PRELIMINARIES ON LLM INFERENCE

We review the KV caching mechanism in transformer-based LLM inference, focusing on the prefill, decoding, and cache eviction phases. Bold capital letters denote matrices, and bold lowercase letters with subscripts denote row vectors. For clarity, we omit the batch, head, and layer indices.

**Prefill Phase.** Let $\boldsymbol{X} \in \mathbb{R}^{l \times d}$ denote the prompt embeddings, and let $\mathbf{Q}, \mathbf{K}, \mathbf{V} \in \mathbb{R}^{l \times d}$ be the corresponding projected query, key, and value matrices, where $l$ is the prompt length and $d$ is the hidden size. The attention output $\mathbf{O} \in \mathbb{R}^{l \times d}$ is computed as

$$\mathbf{O} = \mathbf{A}\mathbf{V}, \quad \mathbf{A} = \text{softmax}(\mathbf{Z}), \quad \mathbf{Z} = \frac{\mathbf{Q}\mathbf{K}^\top}{\sqrt{d}} \in \mathbb{R}^{l \times l},$$

where $\mathbf{Z}$ denotes the pre-softmax attention logits and $\mathbf{A} \in \mathbb{R}^{l \times l}$ is the attention weight matrix. During this phase, the key and value matrices $\mathbf{K}$ and $\mathbf{V}$ are cached to avoid recomputation in subsequent decoding steps.

**Decoding Phase.** At decoding step $t$, let $\boldsymbol{x}_t \in \mathbb{R}^d$ be the newly generated token embedding, and let $\mathbf{q}_t, \mathbf{k}_t, \mathbf{v}_t \in \mathbb{R}^d$ be its projected query, key, and value vectors. After appending $\mathbf{k}_t, \mathbf{v}_t$ to the cache, the key and value matrices extend to shape $s \times d$, where $s = l + t$. The step-$t$ attention output is

$$\mathbf{o}_s = \mathbf{a}_s \boldsymbol{V}, \quad \mathbf{a}_s = \text{softmax}(\mathbf{z}_s), \quad \mathbf{z}_s = \frac{\mathbf{q}_t \mathbf{K}^\top}{\sqrt{d}} \in \mathbb{R}^s.$$

For the remainder of this paper, we use $\mathbf{Q}, \mathbf{K}, \mathbf{V}, \mathbf{A}, \mathbf{O}$ to denote the full matrices corresponding to sequence length $s$, which reduces to prompt length $l$ during the prefill phase when $t = 0$.

**Cache Eviction.** Eviction is triggered when the sequence length $s$ of the KV cache exceeds a predefined budget $N$. After each forward pass, an eviction algorithm selects $s - N$ rows from $\mathbf{K}$ and $\mathbf{V}$ for permanent deletion. By enforcing a fixed cache budget, the model can decode arbitrarily long sequences while maintaining a bounded memory footprint.

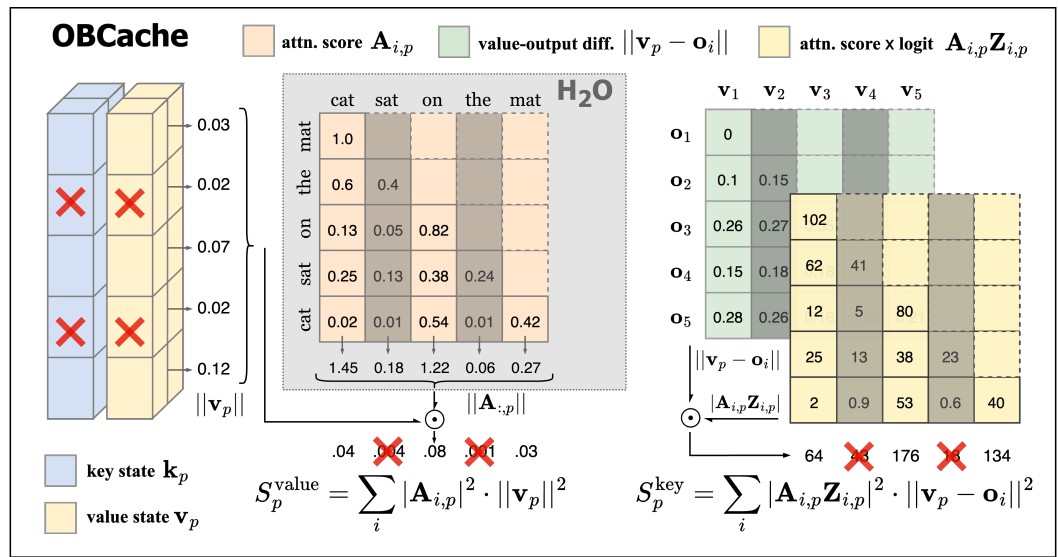

Figure 1: Overview of the OBCACHE scoring mechanism. The diagram shows the eviction process using value-pruning (left) and key-pruning scores (right). Unlike prior methods based solely on attention statistics (gray region), OBCACHE further incorporates value states, attention logits, and outputs to estimate token saliency, explicitly targeting the minimization of eviction-induced errors.

## 3.2 CACHE EVICTION VIA PERTURBATION MINIMIZATION

Cache eviction can be viewed as a form of layer-wise structured pruning, where the saliency of each pruning unit (i.e., a key or a value vector) is quantified by the error it induces in the layer output. However, cache eviction introduces a unique challenge not encountered in classical model pruning: due to the autoregressive nature of generation, the actual error caused by removing a key–value pair at step $s$ affects only future attention outputs, $\mathbf{o}_{s+1}, \mathbf{o}_{s+2}, \ldots$, which are inaccessible at eviction time. We refer to this unobservable quantity as the *true eviction error*.

Although the true eviction error is not directly available, we observe that it can be effectively approximated by measuring perturbations in recent historical attention outputs, specifically $\mathbf{o}_s, \mathbf{o}_{s-1}, \ldots$, when the corresponding KV vectors are pruned. We refer to this measurable surrogate as the *pruning-induced eviction error*, and use it as a proxy objective to estimate token saliency. Building on this insight, we formulate cache eviction as a layer-wise structured pruning problem by treating $\mathbf{V}$ and $\mathbf{K}$ as pruning variables.

**Definition 3.1** (Token Saliency via Pruning-Induced Error). Let $\widehat{\mathbf{V}} = \mathbf{V} + \delta\mathbf{V}$ and $\widehat{\mathbf{K}} = \mathbf{K} + \delta\mathbf{K}$ denote the perturbed value and key matrices after pruning, with resulting perturbed attention weights $\widehat{\mathbf{A}}$ and outputs $\widehat{\mathbf{O}}$. The saliency score of a token position $p$ is defined as the change in recent historical attention outputs $\mathbf{O}$ when $\mathbf{v}_p$ and $\mathbf{k}_p$ are pruned:

$$\boldsymbol{S}_p := \mathcal{L}_{\boldsymbol{e}_p^\top [\widehat{\mathbf{V}}\ \widehat{\mathbf{K}}] = \mathbf{0}}(\widehat{\mathbf{V}}, \widehat{\mathbf{K}}) = f\left(\operatorname{softmax}\left(\frac{\mathbf{Q}\widehat{\mathbf{K}}^\top}{\sqrt{d}}\right)\widehat{\mathbf{V}}\Big|_{\boldsymbol{e}_p^\top [\widehat{\mathbf{V}}\ \widehat{\mathbf{K}}] = \mathbf{0}} - \operatorname{softmax}\left(\frac{\mathbf{Q}\mathbf{K}^\top}{\sqrt{d}}\right)\mathbf{V}\right), \quad (1)$$

where $\boldsymbol{e}_p$ is a unit vector selecting the $p$-th row of $\widehat{\mathbf{V}}$ and $\widehat{\mathbf{K}}$, and $f(\cdot)$ is a norm function.

Following prior works on layer-wise model pruning of LLMs (Frantar & Alistarh, 2023; Sun et al., 2024), we adopt the squared Frobenius norm $\|\cdot\|_F^2$ for its smoothness. To highlight the effectiveness of this pruning-induced error as a proxy objective, we present an empirical example in Section 3.4.

## 3.3 Cache Pruning Scores

Directly recomputing Equation 1 for every position $p$ requires repeated attention operations and is computationally infeasible. Inspired by Optimal Brain Damage (OBD) (LeCun et al., 1989), we approximate it via a second-order Taylor expansion around the unperturbed point $(\mathbf{V}, \mathbf{K})$.

**Theorem 3.2.** *Let $\mathbf{H}_{vv}, \mathbf{H}_{kk}, \mathbf{H}_{vk}$ be the Hessians of $\mathcal{L}$ with respect to $\widehat{\mathbf{V}}, \widehat{\mathbf{K}}$, and their cross-term. The pruning-induced eviction error $\mathcal{L}$ expanded around $(\mathbf{V}, \mathbf{K})$ to the second order is given by:*

$$\mathcal{L}(\widehat{\mathbf{V}}, \widehat{\mathbf{K}}) = \frac{1}{2}\delta\mathbf{V}^\top\mathbf{H}_{vv}\delta\mathbf{V} + \frac{1}{2}\delta\mathbf{K}^\top\mathbf{H}_{kk}\delta\mathbf{K} + \delta\mathbf{V}^\top\mathbf{H}_{vk}\delta\mathbf{K} + \mathcal{O}(\|(\delta\mathbf{V}, \delta\mathbf{K})\|^3). \tag{2}$$

*When $\mathbf{v}_p$ and $\mathbf{k}_p$ are pruned, i.e., $\boldsymbol{e}_p^\top[\widehat{\mathbf{V}}\ \widehat{\mathbf{K}}] = \mathbf{0}$, the above pruning-induced eviction error further simplifies, yielding the saliency score for token position $p$ approximated till second order:*

$$\boldsymbol{S}_p \overset{second}{\underset{order}{=}} \frac{1}{2}\mathbf{v}_p^\top[\mathbf{H}_{vv}]_{pp}\mathbf{v}_p + \frac{1}{2}\mathbf{k}_p^\top[\mathbf{H}_{kk}]_{pp}\mathbf{k}_p + \mathbf{v}_p^\top[\mathbf{H}_{vk}]_{pp}\mathbf{k}_p, \tag{3}$$

In Equation 2, the first-order terms vanish at the expansion point since $\widehat{\mathbf{O}} - \mathbf{O} = \mathbf{0}$. The simplification to Equation 3 uses the fact that the off-diagonal blocks of $\mathbf{H}_{vv}, \mathbf{H}_{kk}, \mathbf{H}_{vk}$ do not contribute to $\mathcal{L}$. This mirrors the diagonal assumption adopted in OBD. Full proof can be seen in Appendix B.

By explicitly evaluating the Hessian sub-blocks $[\mathbf{H}_{vv}]_{pp}$, $[\mathbf{H}_{kk}]_{pp}$, and $[\mathbf{H}_{vk}]_{pp}$, we next derive closed-form eviction scores that are both interpretable and efficient to compute. Currently, Equation 3 captures the joint impact of perturbing both $\mathbf{V}$ and $\mathbf{K}$. We also consider simplified variants where either $\mathbf{V}$ or $\mathbf{K}$ is perturbed independently while the other remains fixed. As a result, OBCACHE features three output-aware saliency scores, as demonstrated in Equations 4–6 below.

**Proposition 3.3** (Value-Pruning Score). *When only $\mathbf{V}$ is the pruning unit, i.e., $\boldsymbol{e}_p^\top\widehat{\mathbf{V}} = \mathbf{0}$, the pruning-induced eviction error reduces to the first term in Equation 3, which is given by:*

$$\boldsymbol{S}_p^{\text{value}} = \frac{1}{2}\mathbf{v}_p^\top[\mathbf{H}_{vv}]_{pp}\mathbf{v}_p = \sum_i |\mathbf{A}_{i,p}|^2 \cdot \|\mathbf{v}_p\|^2. \tag{4}$$

This score corresponds to the squared $L_2$-norm of the $p$-th column of the attention weight matrix, scaled by the squared $L_2$-norm of the value vector $\mathbf{v}_p$. Its full proof including the derivation of the Hessian sub-block $[\mathbf{H}_{vv}]_{pp}$ can be seen in Appendix B.2.

**Proposition 3.4** (Key-Pruning Score). *When only $\mathbf{K}$ is the pruning unit, i.e., $\boldsymbol{e}_p^\top\widehat{\mathbf{K}} = \mathbf{0}$, the pruning-induced eviction error reduces to the second term in Equation 3, which is given by:*

$$\boldsymbol{S}_p^{\text{key}} = \frac{1}{2}\delta\mathbf{K}^\top\mathbf{H}_{kk}\delta\mathbf{K} = \sum_i |\mathbf{A}_{i,p} \cdot \mathbf{Z}_{i,p}|^2 \cdot \|\mathbf{v}_p - \mathbf{o}_i\|^2. \tag{5}$$

This score captures the deviation between the value vector and attention output, weighted by both the attention weights and the pre-softmax logits. Key pruning generally incurs larger errors than value pruning, as it alters the entire attention distribution. Its proof can be seen in Appendix B.3.

**Proposition 3.5** (Joint-Pruning Score). *When $\mathbf{V}$ and $\mathbf{K}$ are treated as a combined pruning unit, the pruning-induced eviction error, as given by Equation 3, has the form:*

$$\boldsymbol{S}_p^{\text{joint}} = 2\sum_i |\mathbf{A}_{i,p}|^2 \cdot \mathbf{Z}_{i,p} \cdot (\|\mathbf{v}_p\|^2 - \mathbf{v}_p^\top\boldsymbol{o}_i) + \boldsymbol{S}_p^{value} + \boldsymbol{S}_p^{key}. \tag{6}$$

This score captures both the individual and interactive effects of pruning the key and value states, providing the most comprehensive estimate of the pruning-induced eviction error. The derivation of the cross-term can be seen in Appendix B.4.

We note that OBCACHE scores can be applied for both prefill and decoding. In the prefill phase, the saliency score $\boldsymbol{S}_p$ can be used to greedily evict multiple tokens to achieve a desired sparsity in a one-shot manner. During decoding, by accumulating $\boldsymbol{S}_p$ over time, OBCACHE can support real-time updates to token-wise saliency, enabling dynamic KV cache eviction as generation progresses. For models using grouped-query attentions, we include an additional score derivation in Appendix B.5.

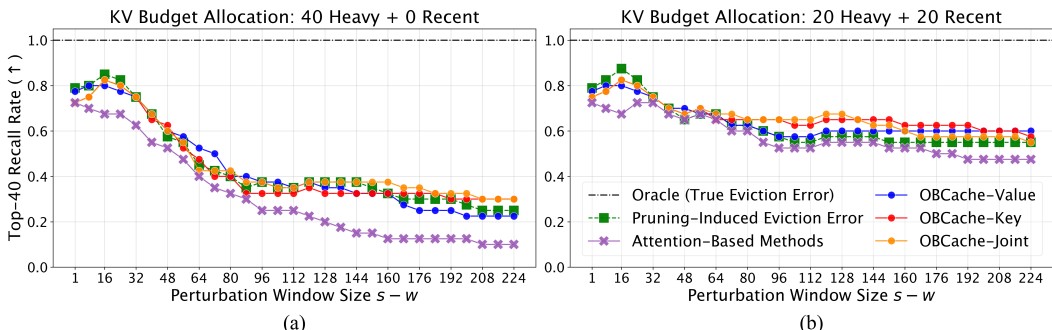

Figure 2: Recall rate of the top-40 salient tokens identified by the oracle eviction error. Results are collected from a 4K-context passkey retrieval task using LLaMA-3.2-3B-Instruct. The left plot (a) shows recall of top-40 tokens selected by different scoring methods. The right plot (b) demonstrates how allocating a fixed recent window improves oracle recall when perturbation windows are large, mitigating the structural bias disproportionately favoring earlier tokens.

### 3.4 CONNECTION TO EXISTING METHODS

Our framework naturally recovers existing attention-based eviction strategies as special cases. To show this, we introduce an additional index $w \in [1, s]$ and use $\mathbf{A}_{w:s}$ to denote the rows of the attention matrix $\mathbf{A}$ corresponding to the query positions from $w$ to $s$. Consider an alternative formulation that minimizes perturbations not in the attention outputs $\mathbf{O}$, but in the historical attention rows $\mathbf{A}_{w:s}$. In this case, the pruning-induced error reduces to:

$$\boldsymbol{S}_p^{\text{attn-based}} := \mathcal{L}_{\widehat{\mathbf{A}}_{:,p}=\mathbf{0}}(\widehat{\mathbf{A}}) = \left\| \widehat{\mathbf{A}}_{w:s} \Big|_{\widehat{\mathbf{A}}_{:,p}=\mathbf{0}} - \mathbf{A}_{w:s} \right\|_{1,1} \tag{7}$$

where the pruning unit also simplifies to an attention matrix column. We refer to query positions from $w$ to $s$ as the *perturbation window*. Equation 7 can be directly simplified to the form:

$$\boldsymbol{S}_p^{\text{attn-based}} = \sum_{i=w}^{s} |\mathbf{A}_{i,p}|, \tag{8}$$

which corresponds to an $L_1$-norm variant of our value-pruning score $\boldsymbol{S}_p^{\text{value}}$, but without incorporating any value-state information. This formulation connects directly to several recent methods. Specifically, $H_2O$ (Zhang et al., 2023) sets $w = 1$, accumulating attention weights over the entire sequence history. TOVA (Oren et al., 2024) sets $w = s$, targeting only the most recent query position. SnapKV (Li et al., 2024) adopts a short window (i.e., $w \gg 1$), emphasizing recent attentions. By varying the choice of $w$, these methods effectively target different perturbation windows.

In OBCACHE, when we also relax the objective to the output error within the perturbation window:

$$\boldsymbol{S}_p = \left\| \text{softmax}\left(\frac{\mathbf{Q}_{w:s}\widehat{\mathbf{K}}^\top}{\sqrt{d}}\right)\widehat{\mathbf{V}} \Big|_{\boldsymbol{e}_p^\top[\widehat{\mathbf{V}} \ \widehat{\mathbf{K}}]=\mathbf{0}} - \text{softmax}\left(\frac{\mathbf{Q}_{w:s}\mathbf{K}^\top}{\sqrt{d}}\right)\mathbf{V} \right\|_F^2, \tag{9}$$

the resulting scores also become localized to the same set of query positions $w$ through $s$. Therefore, OBCACHE generalizes these prior approaches by further introducing output-aware signals, enabling more informed eviction decisions that go beyond raw attention statistics.

### 3.5 EFFECTIVENESS OF PRUNING-INDUCED EVICTION ERROR

To further support our pruning-based formulation for cache eviction, we present a qualitative analysis on a passkey retrieval task, as shown in Figure 2. We begin by establishing an oracle baseline based on the *true eviction error*, which is measured as the perturbation in the first decoding-step output $\mathbf{o}_{l+1}$ caused by evicting each cached token during the prefill phase. The top-$k$ tokens with the largest oracle errors are treated as ground-truth important positions.

To evaluate the effectiveness of the *pruning-induced eviction error* as a proxy, we compute Equation 1 exactly for each candidate position in the prefill phase and select the top-$k$ accordingly. As

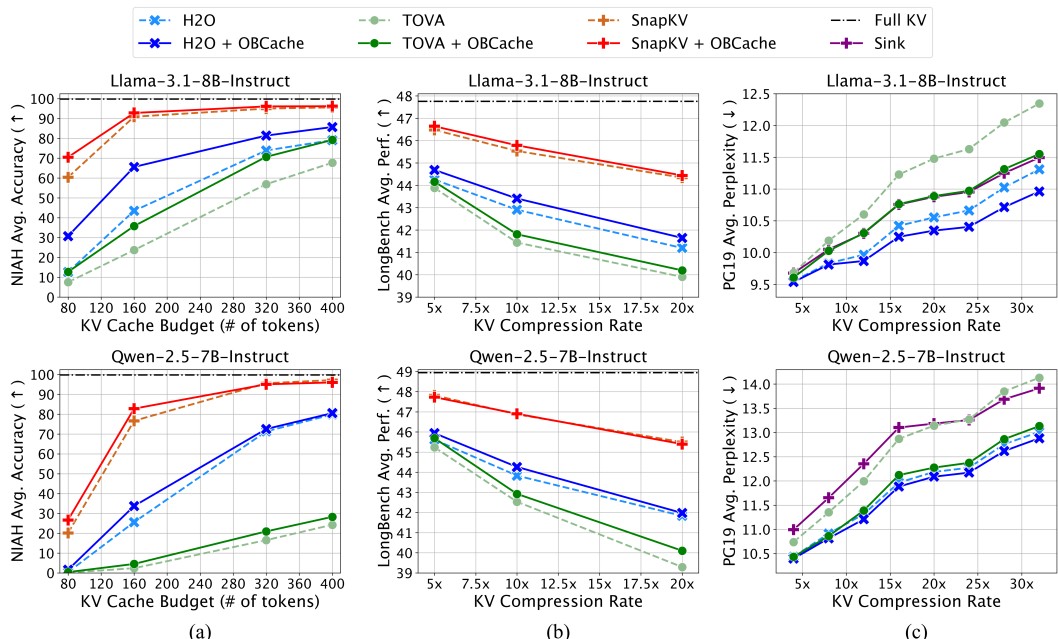

Figure 3: Overall long-context performance evaluation of LLaMA and Qwen. When integrated with OBCACHE scores, existing cache eviction baselines achieve superior compression and performance trade-off on the Needle-In-A-Haystack (a), LongBench (b), and perplexity (c) benchmarks.

shown in the figure, when the perturbation window is appropriately chosen, the proxy achieves up to 85% recall of the oracle top-$k$ selections. Next, we demonstrate the superiority of OBCACHE. Although derived via second-order Taylor approximation, OBCACHE scores achieve nearly identical ranking performance to the exact proxy, while being significantly more efficient due to their closed-form expression. Compared to attention-based methods, our scores consistently yield higher oracle recall, showcasing the benefit of incorporating output-aware signals into saliency estimation.

We observe that recall degrades when the perturbation window becomes larger. This issue stems from attention causality: earlier tokens attend to more queries and thus accumulate disproportionately higher saliency scores, introducing a structural bias that favors retaining initial tokens. To mitigate this, $H_2O$ reserves a portion of the cache budget for a recent window, ensuring that the most recent tokens are never evicted. This policy is also complementary to OBCACHE, and we find that recall further improves when a fixed 20-token window is reserved, as demonstrated in Figure 2b.

## 4 EXPERIMENTS

In this section, we conduct comprehensive experiments to evaluate the effectiveness of OBCACHE in enhancing existing score-based cache eviction methods, thereby improving performance across a range of long-context benchmarks. We evaluate OBCACHE scores for both static cache eviction in the prefill phase (Section 4.2) and dynamic cache eviction in the decoding phase (Section 4.3). Our evaluation demonstrates that OBCACHE consistently achieves a superior trade-off between compression and performance. Additionally, we conduct efficiency experiments to showcase the negligible overhead introduced by our scoring mechanism, which is detailed in Appendix D.1.

### 4.1 EXPERIMENTAL SETUP

**Datasets.** For prefill-phase cache eviction, we adopt two widely recognized long-context benchmarks: Needle-in-a-Haystack (NIAH) (Kamradt, 2023) and LongBench (Bai et al., 2023b). NIAH tests a model's ability to retrieve a small but crucial "needle" randomly embedded in a lengthy document. Our implementation follows the RULER benchmark (Hsieh et al., 2024) and we evaluate across multiple context lengths (from 4K to 32K) under varying cache budgets. LongBench

Table 1: Results of the Needle-in-a-Haystack test on LLaMA-3.1-8B-Instruct. Each accuracy value corresponds to exact match over 250 random samples drawn from the RULER benchmark.

| KV Budget (# tokens) | 4K Context | | | | 8K Context | | | | 16K Context | | | | 32K Context | | | | Avg. |
|---|---|---|---|---|---|---|---|---|---|---|---|---|---|---|---|---|---|
| | 80 | 160 | 320 | 400 | 80 | 160 | 320 | 400 | 80 | 160 | 320 | 400 | 80 | 160 | 320 | 400 | |
| H2O | 7.6 | 36.8 | 68.0 | 77.2 | 10.8 | 47.2 | 75.2 | 79.2 | 14.8 | 45.2 | 80.0 | 82.8 | 17.2 | 44.8 | 72.4 | 77.2 | 52.28 |
| + OBCACHE-VALUE | 15.2 | 54.4 | 76.8 | 82.4 | 21.6 | 60.4 | 82.4 | 85.2 | 30.8 | 72.0 | 84.4 | 85.6 | 32.4 | 61.6 | 82.4 | 86.4 | 63.38 |
| + OBCACHE-KEY | 19.2 | 55.6 | 77.2 | 84.0 | 29.6 | 60.4 | 82.8 | 86.8 | 39.6 | 70.8 | 83.6 | 84.8 | 35.2 | 65.2 | 84.0 | 88.0 | 65.42 |
| + OBCACHE-JOINT | 25.6 | 62.0 | 77.6 | 82.8 | 33.2 | 67.2 | 83.2 | 88.0 | 34.4 | 72.0 | 82.0 | 84.8 | 29.6 | 61.2 | 82.8 | 87.2 | **65.85** |
| TOVA | 7.2 | 24.4 | 63.6 | 76.0 | 9.2 | 26.4 | 62.4 | 68.8 | 6.8 | 21.6 | 55.6 | 68.0 | 6.8 | 22.4 | 46.0 | 58.0 | 38.95 |
| + OBCACHE-VALUE | 12.4 | 34.4 | 75.6 | 84.0 | 10.4 | 37.6 | 71.6 | 81.2 | 10.0 | 34.4 | 67.6 | 77.2 | 14.4 | 31.6 | 61.2 | 68.4 | 48.25 |
| + OBCACHE-KEY | 12.8 | 37.2 | 78.0 | 86.4 | 12.4 | 38.0 | 73.6 | 82.8 | 12.0 | 34.8 | 69.2 | 79.2 | 15.2 | 34.0 | 60.4 | 68.4 | **49.65** |
| + OBCACHE-JOINT | 10.4 | 34.0 | 75.2 | 85.2 | 12.8 | 39.2 | 74.4 | 80.4 | 12.8 | 37.6 | 69.6 | 78.8 | 14.8 | 32.4 | 63.6 | 72.4 | 49.60 |
| SnapKV | 66.4 | 84.8 | 92.8 | 93.2 | 58.4 | 92.0 | 96.4 | 97.2 | 55.6 | 90.8 | 94.0 | 95.2 | 61.2 | 96.0 | 96.8 | 96.8 | 85.48 |
| + OBCACHE-VALUE | 64.4 | 83.6 | 92.4 | 92.8 | 63.6 | 92.4 | 96.8 | 97.2 | 60.0 | 89.2 | 93.6 | 94.4 | 64.0 | 93.2 | 96.8 | 97.2 | 85.72 |
| + OBCACHE-KEY | 73.6 | 82.8 | 92.4 | 92.4 | 72.8 | 92.4 | 97.2 | 96.4 | 70.8 | 93.6 | 96.0 | 96.0 | 74.0 | 96.8 | 96.8 | 97.2 | 88.82 |
| + OBCACHE-JOINT | 68.4 | 86.4 | 92.0 | 92.8 | 72.0 | 95.2 | 97.6 | 96.8 | 73.6 | 95.6 | 96.8 | 97.2 | 68.0 | 94.0 | 98.0 | 98.0 | **88.90** |

evaluates LLM's long-context understanding via 16 datasets spanning six task categories: single-document QA, multi-document QA, summarization, few-shot learning, synthetic reasoning, and code completion. The average input length across all datasets is 6,711 words, making KV cache optimization critical during inference. For decoding-phase cache eviction, we follow the setup in StreamingLLM (Xiao et al., 2024), and report language modeling perplexity on PG19 (**?**), a dataset of 100 books with an average length of 70K tokens.

**Baselines.** We compare OBCACHE with three state-of-the-art cache eviction methods: $H_2O$ (Zhang et al., 2023), TOVA (Oren et al., 2024), and SnapKV (Li et al., 2024), all of which rely solely on attention statistics but differ in eviction strategy. $H_2O$ accumulates historical attention weights while retaining a fixed window of recent tokens. TOVA selects tokens based solely on the most recent query's attention weights and does not require a recent window. SnapKV, designed for prefill-only eviction, scores tokens based on a recent attention window and applies pooling to smooth token importance, making it effective for retrieval-centric tasks. We integrate OBCACHE into these baselines by replacing their attention-based scores with our perturbation-aware scores while preserving their original eviction strategies. We refer to our eviction methods using the value-pruning score (Equation 4) as OBCACHE-VALUE, the key-pruning score (Equation 5) as OBCACHE-KEY, and joint pruning score (Equation 6) as OBCACHE-JOINT, respectively.

**Implementation Details.** We use LLaMA-3.1-8B-Instruct (Grattafiori et al., 2024) and Qwen-2.5-7B-Instruct (Bai et al., 2023a) as our backbone LLMs, both of which natively supports 128K context windows. All experiments are implemented using the HuggingFace Transformers library (Wolf et al., 2020) with PyTorch (Paszke et al., 2019). To improve efficiency, we modify existing baseline's prefill-phase attention operation using FlashAttention-2 (Dao, 2023), which reduces memory overhead for long-context samples. For prefill-phase eviction, we follow setups in SnapKV, where cache eviction occurs only once before decoding begins. This reflects real-world scenarios where prefill KV cache dominates memory usage compared to the ones in decoding. For decoding-phase eviction, we follow $H_2O$ to fix a recent window, while evicting tokens at every step to evaluate the dynamic impact of different saliency scores. Full implementation details are provided in Appendix C.

## 4.2 PREFILL CACHE EVICTION

**Needle-In-A-Haystack Results.** Table 1 presents detailed NIAH passkey retrieval accuracy under different cache budgets and context lengths, comparing baseline eviction methods with their OBCACHE-enhanced counterparts. In addition, we aggregate accuracies over all context sizes and present the compression-performance trade-off curves for OBCACHE-JOINT in Figure 3a.

Across all baselines, compression rates, and context lengths, integrating OBCACHE scores consistently improves the retrieval accuracy. In particular, when applied to $H_2O$ and TOVA, OBCACHE yields substantial accuracy gains, achieving over $1.26\times$ the original accuracy. These increments are more significant under extreme compression rates, highlighting the advantage of our output-aware saliency in identifying critical tokens. For SnapKV, the improvement is relatively smaller (3% ab-

solute increase) as it already performs near saturation. However, under severe compression settings (e.g., 80-token budgets), OBCACHE still achieves 10% absolute gains as depicted in Figure 3a. These results demonstrate the robustness of OBCACHE scores in constrained memory settings. Overall, all three baselines benefit from the integration of OBCACHE at every tested compression rate, which highlights the advantage of our output-aware scoring mechanism over attention-based heuristics.

In addition to the comparison with baselines, Table 1 also provides an ablation on OBCACHE's three score variants, derived via different pruning unit assumptions. We observe that eviction scores derived from key pruning, i.e., OBCACHE-KEY and OBCACHE-JOINT, tend to outperform the value-only variant OBCACHE-VALUE. This aligns with our expectation that pruning keys has greater impact, due to their role in shaping attention distributions and ultimately the model predictions. Accounting for key sensitivity thus yields more accurate saliency estimates and better performance. Although the OBCACHE-VALUE variant underperforms slightly, it remains appealing for its simplicity: it requires only an additional scaling factor based on value-state norms, making it nearly as efficient as baseline attention scores while still delivering substantial performance gains.

**LongBench Results.** We report the full table of results across all 16 LongBench datasets when retaining only 5%, 10%, 20% of the KV tokens in Appendix D.3. To highlight the benefits of OBCACHE, we also present the average task performance across different compression rates in Figure 3b. For both LLaMA-3.1 and Qwen-2.5 models, we observe consistent improvements in average performance when integrating OBCACHE into existing baselines. Interestingly, the improvement for SnapKV on Qwen is relatively smaller. We hypothesize that this is due to SnapKV's pooling-based smoothing mechanism providing limited additional benefit when applied to our key-pruning scores, which already capture more nuanced token saliency beyond raw attention statistics. Overall, the results confirm that replacing attention-only scores with OBCACHE saliency enables more effective preservation of task-critical tokens under various compression levels.

### 4.3 DYNAMIC CACHE EVICTION

Beyond static prefill-phase cache eviction, OBCACHE is also applicable to decoding-time scenarios, where tokens must be evicted dynamically. To evaluate this, we measure language modeling perplexity at varying sequence lengths on the PG19 test set, using a fixed cache budget of 1024 tokens. Following Xiao et al. (2024), we allocate 4 fixed initial tokens for all methods, as these tokens act as attention sinks and are crucial for long-sequence generation quality. As shown in Figure 4, the Sink baseline, which statically allocates the cache to fixed initial and recent tokens, demonstrates the highest perplexity. The H2O baseline, which dynamically selects important tokens based on cumulative attention statistics, performs moderately better. In contrast, all OBCACHE variants, which also accumulate output-aware scores over time, consistently outperform H2O across all sequence lengths. As observed, OBCACHE-JOINT does not yield better performance than OBCACHE-KEY, suggesting that a better combination of key and value pruning scores may exist beyond the current plain additive formulation. SnapKV is not evaluated as it does not support decoding-phase eviction. We also present the perplexity-compression trade-off curves in Figure 3c, where baselines integrated with OBCACHE scores all exhibit consistently lower perplexity at all compression levels.

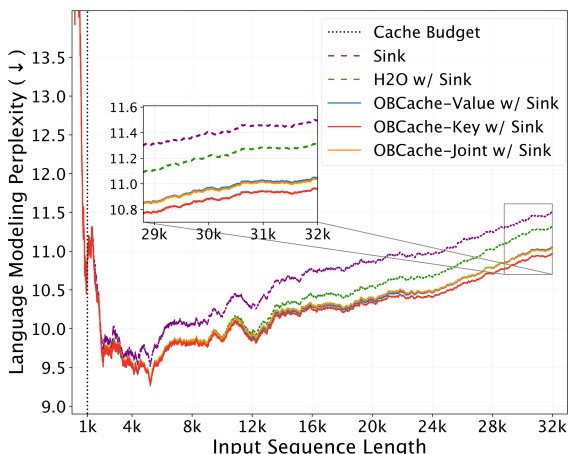

Figure 4: Language modeling perplexity evaluation on PG19. We prompt Llama-3.1-8B-Instruct with 1 to 32K tokens and measure the perplexity of output tokens at varying context lengths. The KV cache budget for all methods is fixed at 1024 number of tokens.

These results confirm that our perturbation-aware saliency scores can improve memory utilization and preserve long-range dependencies critical for continuous generation.

## 5 CONCLUSION

In this work, we introduce OBCACHE, a principled framework for KV cache eviction in large language model inference, grounded in a perturbation minimization perspective. By casting cache eviction as a layer-wise structured pruning problem, we derived token saliency scores that aim to minimize the impact of removal on attention outputs. Leveraging second-order Taylor approximations, OBCACHE yields efficient, closed-form token saliency scores that generalize existing attention-based heuristics while incorporating more effective output-aware signals. Extensive experiments across both prefilling and decoding scenarios on long-context benchmarks demonstrate that OB-CACHE consistently outperforms state-of-the-art baselines, including $H_2O$, TOVA, and SnapKV, achieving superior performance-compression trade-offs, particularly under tight memory budgets. OBCACHE also offers a flexible foundation for principled KV cache compression. Its structured pruning framework can be extended to other settings by modifying the pruning objective or unit. For instance, one could adapt OBCACHE to channel-wise KV pruning, or explore cache merging strategies by relaxing the diagonal approximation in our analysis. These directions open promising paths towards more effective and theoretically grounded KV cache management in long-context LLMs.

## ETHICS STATEMENT

The authors of this work have read and commit to adhering to the Code of Ethics. Our research introduces OBCache, a principled framework for KV cache eviction in large language models, formulated as a structured pruning problem. To the best of our knowledge, this work does not raise direct ethical concerns. It does not involve the use of personally identifiable information, sensitive human-subject data, or applications with immediate potential for societal harm.

## REPRODUCIBILITY STATEMENT

We provide the complete source code in the supplementary materials. Further details on the experimental setup, including hyperparameters, datasets, and model architecture, are documented in the Appendix.

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

# Appendix

## A DISCUSSION

### A.1 LIMITATIONS AND FUTURE WORKS

While effective, OBCACHE presents several opportunities for further improvement. One notable limitation is the structural bias towards earlier tokens, a common drawback in many existing cache eviction methods, which also emerges in OBCACHE, particularly as the perturbation window increase. Although heuristics like retaining a fixed-size recent window help mitigate this bias by leveraging the locality of attention patterns, such strategies remain static and empirically driven. Future work should more systematically investigate the design and adaptability of perturbation windows to better improve saliency estimation. Additionally, OBCACHE's current dynamic cache eviction strategy in the decoding phase directly follows $H_2O$ by accumulating saliency scores over time, effectively minimizing perturbations over the full sequence history. However, under our formulation, alternative objectives could be explored. For example, accumulating scores only within a recent attention window. Such variants may yield improved adaptability in long-context generation.

On the theoretical side, OBCACHE introduces a flexible, perturbation-minimization framework that can be extended to a broader class of KV cache compression strategies. For instance, due to the inherent flexibility of structured pruning, one could adapt the OBCACHE formulation to key channel pruning by redefining the pruning units from tokens to channels. Leveraging approximation techniques, it is possible to derive corresponding closed-form saliency scores from an output-aware perspective, enabling informed channel-level pruning decisions. Moreover, our approach relies on the diagonal Hessian assumption used in Optimal Brain Damage, which supports only token removal. In contrast, another classical pruning theory Optimal Brain Surgeon relaxes this assumption by allowing compensation across pruning units, adjusting remaining parameters to reduce loss. This perspective aligns naturally with recent works on cache merging, and could inspire more theoretically grounded cache management techniques for long-context LLMs.

Overall, OBCACHE offers a flexible foundation for future research into KV cache compression. Its structured and theoretically motivated formulation opens up promising directions for designing more effective and principled KV cache management.

### A.2 USE OF LARGE LANGUAGE MODELS

In this work, we used large language models (LLMs) to assist with manuscript editing. LLMs were used to help polish the language of the manuscript. This includes surface-level edits such as improving clarity, grammar, and conciseness of English expressions. All technical content, algorithmic designs, and empirical results were authored and validated by the authors. No part of the scientific contributions was generated by or delegated to an LLM.

## B THEORETICAL ANALYSIS

In this section, we demonstrate the full proof and derivations to obtain the OBCACHE scores in Equation 4, Equation 5 and Equation 6. We use consistent notations as previously defined in Section 3.1.

### B.1 OBCACHE OBJECTIVE FUNCTION

As demonstrated in Section 3.2 and Section 3.4, we formulate the saliency score for a candidate key-value token as Equation 1, which is the *pruning-induced eviction error* within a *perturbation window* for query positions $w$ through $s$:

$$\mathcal{L}(\widehat{\mathbf{V}}, \widehat{\mathbf{K}}) := \left\| \widehat{\mathbf{O}}_{w:s} - \mathbf{O}_{w:s} \right\|_F^2 = \left\| \text{softmax}\left(\frac{\mathbf{Q}_{w:s}\widehat{\mathbf{K}}^\top}{\sqrt{d}}\right)\widehat{\mathbf{V}} - \text{softmax}\left(\frac{\mathbf{Q}_{w:s}\mathbf{K}^\top}{\sqrt{d}}\right)\mathbf{V} \right\|_F^2. \quad (10)$$

In what follows, we let $\mathbf{O}_{i,j} = \mathbf{a}_i \mathbf{V}_{:,j}$ denote the $j$-th feature element in the $i$-th row vector of the attention output matrix $\mathbf{O}$, where $\mathbf{a}_i$ is the $i$-th row vector of the attention weight matrix. The

squared Frobenius norm objective in Equation 10 can be explicitly decomposed into a summation form across all element-wise squared error:

$$\mathcal{L}(\widehat{\mathbf{V}}, \widehat{\mathbf{K}}) = \sum_{i=w}^{s} \sum_{j=1}^{d} |\widehat{\mathbf{O}}_{i,j} - \mathbf{O}_{i,j}|^2$$

$$= \sum_{i=w}^{s} \sum_{j=1}^{d} |\mathrm{softmax}\big(\tfrac{\mathbf{q}_i \widehat{\mathbf{K}}^\top}{\sqrt{d}}\big)\widehat{\mathbf{V}}_{:,j} - \mathrm{softmax}\big(\tfrac{\mathbf{q}_i \mathbf{K}^\top}{\sqrt{d}}\big)\mathbf{V}_{:,j}|^2 \qquad (11)$$

$$\triangleq \sum_{i=w}^{s} \sum_{j=1}^{d} \boldsymbol{E}_{i,j}.$$

For clarity in the following derivations, we define $\boldsymbol{E}_{i,j} := |\widehat{\mathbf{O}}_{i,j} - \mathbf{O}_{i,j}|^2$ as the squared difference of the output element $\mathbf{O}_{i,j}$ before and after applying a perturbation $\delta\mathbf{V}$ and $\delta\mathbf{K}$. Expanding explicitly, the *element-wise output perturbation* for $\mathbf{O}_{i,j}$ is:

$$\boldsymbol{E}_{i,j}(\widehat{\mathbf{V}}_{:,j}, \widehat{\mathbf{K}}) = |\mathrm{softmax}\big(\tfrac{\mathbf{q}_i \widehat{\mathbf{K}}^\top}{\sqrt{d}}\big)\widehat{\mathbf{V}}_{:,j} - \mathrm{softmax}\big(\tfrac{\mathbf{q}_i \mathbf{K}^\top}{\sqrt{d}}\big)\mathbf{V}_{:,j}|^2. \qquad (12)$$

Given this element-wise objective, we follow the Optimal Brain Damage (OBD) theory (LeCun et al., 1989) to apply a second-order Taylor expansion of $\boldsymbol{E}_{i,j}$ around $(\mathbf{V}_{:,j}, \mathbf{K})$, and analytically approximate the element-wise output perturbation:

$$\boldsymbol{E}_{i,j}(\widehat{\mathbf{V}}_{:,j}, \widehat{\mathbf{K}}) \stackrel{\mathrm{second}}{\underset{\mathrm{order}}{=}} \boldsymbol{E}_{i,j}(\mathbf{V}_{:,j}, \mathbf{K}) + \delta\mathbf{V}_{:,j}^\top \frac{\partial \boldsymbol{E}_{i,j}(\mathbf{V}_{:,j}, \mathbf{K})}{\partial \widehat{\mathbf{V}}_{:,j}} + \frac{1}{2}\delta\mathbf{V}_{:,j}^\top \frac{\partial^2 \boldsymbol{E}_{i,j}(\mathbf{V}_{:,j}, \mathbf{K})}{\partial \widehat{\mathbf{V}}_{:,j}^2} \delta\mathbf{V}_{:,j}$$

$$+ \delta\mathbf{K}^\top \frac{\partial \boldsymbol{E}_{i,j}(\mathbf{V}_{:,j}, \mathbf{K})}{\partial \widehat{\mathbf{K}}} + \frac{1}{2}\delta\mathbf{K}^\top \frac{\partial^2 \boldsymbol{E}_{i,j}(\mathbf{V}_{:,j}, \mathbf{K})}{\partial \widehat{\mathbf{K}}^2} \delta\mathbf{K}$$

$$+ \delta\mathbf{V}_{:,j}^\top \frac{\partial^2 \boldsymbol{E}_{i,j}(\mathbf{V}_{:,j}, \mathbf{K})}{\partial \widehat{\mathbf{V}}_{:,j} \partial \widehat{\mathbf{K}}} \delta\mathbf{K} + \mathcal{O}(\|(\delta\mathbf{V}_{:,j}, \delta\mathbf{K})\|^3).$$

Through substitution into Equation 12, the constant term $\boldsymbol{E}_{i,j}(\mathbf{V}_{:,j}, \mathbf{K})$ vanishes because we have $\widehat{\mathbf{O}}_{i,j} - \mathbf{O}_{i,j} = 0$ at the expansion point $(\mathbf{V}_{:,j}, \mathbf{K})$. Therefore, the full matrix-wise objective, *pruning-induced eviction error*, approximated via a Taylor-series to the second order, becomes:

$$\mathcal{L}(\widehat{\mathbf{V}}, \widehat{\mathbf{K}}) \stackrel{\mathrm{second}}{\underset{\mathrm{order}}{=}} \sum_{i=w}^{s} \sum_{j=1}^{d} \delta\mathbf{V}_{:,j}^\top \frac{\partial \boldsymbol{E}_{i,j}(\mathbf{V}_{:,j}, \mathbf{K})}{\partial \widehat{\mathbf{V}}_{:,j}} + \frac{1}{2}\delta\mathbf{V}_{:,j}^\top \frac{\partial^2 \boldsymbol{E}_{i,j}(\mathbf{V}_{:,j}, \mathbf{K})}{\partial \widehat{\mathbf{V}}_{:,j}^2} \delta\mathbf{V}_{:,j} \qquad (13)$$

$$\sum_{i=w}^{s} \sum_{j=1}^{d} \delta\mathbf{K}^\top \frac{\partial \boldsymbol{E}_{i,j}(\mathbf{V}_{:,j}, \mathbf{K})}{\partial \widehat{\mathbf{K}}} + \frac{1}{2}\delta\mathbf{K}^\top \frac{\partial^2 \boldsymbol{E}_{i,j}(\mathbf{V}_{:,j}, \mathbf{K})}{\partial \widehat{\mathbf{K}}^2} \qquad (14)$$

$$\sum_{i=w}^{s} \sum_{j=1}^{d} \delta\mathbf{V}_{:,j}^\top \frac{\partial^2 \boldsymbol{E}_{i,j}(\mathbf{V}_{:,j}, \mathbf{K})}{\partial \widehat{\mathbf{V}}_{:,j} \partial \widehat{\mathbf{K}}} \delta\mathbf{K}. \qquad (15)$$

## B.2 Isolated Value-Pruning Score

In isolated value pruning, the key-cache matrix is not perturbed and is considered a constant not affecting $\mathcal{L}$. Therefore, minimizing the pruning-induced eviction error reduces to minimizing Equation 13. To derive saliency scores in closed form, we begin by deriving expressions for the gradient and Hessian with respect to $\widehat{\mathbf{V}}_{:,j}$. The first-order derivative of $\boldsymbol{E}_{i,j}$ with respect to $\widehat{\mathbf{V}}_{:,j}$ is:

$$\frac{\partial \boldsymbol{E}_{i,j}}{\partial \widehat{\mathbf{V}}_{:,j}} = \frac{\partial}{\partial \widehat{\mathbf{V}}_{:,j}}|\widehat{\mathbf{O}}_{i,j} - \mathbf{O}_{i,j}|^2 = 2(\widehat{\mathbf{O}}_{i,j} - \mathbf{O}_{i,j})\frac{\partial}{\partial \widehat{\mathbf{V}}_{:,j}}\hat{\mathbf{a}}_i \widehat{\mathbf{V}}_{:,j} = 2(\widehat{\mathbf{O}}_{i,j} - \mathbf{O}_{i,j})\hat{\mathbf{a}}_i. \qquad (16)$$

When evaluated at $(\mathbf{V}_{:,j}, \mathbf{K})$, we again have $\widehat{\mathbf{O}}_{i,j} - \mathbf{O}_{i,j} = 0$, so the gradient term vanishes to zero. The Hessian of $\mathcal{L}$ with respect to $\widehat{\mathbf{V}}_{:,j}$ evaluated at $(\mathbf{V}_{:,j}, \mathbf{K})$ is:

$$\frac{\partial^2 \boldsymbol{E}_{i,j}(\mathbf{V}_{:,j}, \mathbf{K})}{\partial \widehat{\mathbf{V}}_{:,j}^2} = 2\frac{\partial}{\partial \widehat{\mathbf{O}}_{i,j}}\Big((\widehat{\mathbf{O}}_{i,j} - \mathbf{O}_{i,j})\hat{\mathbf{a}}_i^\top\Big)\Big(\frac{\partial \widehat{\mathbf{O}}_{i,j}}{\partial \widehat{\mathbf{V}}_{:,j}}\Big)^\top \Big|_{(\mathbf{V}_{:,j}, \mathbf{K})} = 2\mathbf{a}_i^\top \mathbf{a}_i. \qquad (17)$$

Substituting the gradient and Hessian back into Equation 13, we get the pruning-induced eviction error when value states are considered as isolate pruning units:

$$\mathcal{L}^{\text{value}} = \sum_{i=w}^{s} \sum_{j=1}^{d} \delta \mathbf{V}_{:,j}^{\top} (\mathbf{a}_i^{\top} \mathbf{a}_i) \delta \mathbf{V}_{:,j} = \sum_{i=w}^{s} \sum_{j=1}^{d} |\mathbf{a}_i \delta \mathbf{V}_{:,j}|^2. \tag{18}$$

Next, the row-wise pruning constraint $e_p^{\top} \widehat{\mathbf{V}} = \mathbf{0}$ implies that when pruning the $p$-th token position, the value cache perturbation $\delta \mathbf{V}_{:,j}$ is explicitly in the form:

$$\delta \mathbf{V}_{t,j} = \begin{cases} -\mathbf{V}_{p,j}, & \text{when } t = p \\ 0, & \text{when } t \neq p \end{cases}, \quad \forall \, j = 1, ..., d.$$

Substituting this value perturbation into $\mathcal{L}^{\text{value}}$, we obtain the output perturbation when pruning the $p$-th value vector from the value cache $\mathbf{V}$:

$$\boldsymbol{S}_p^{\text{value}} = \sum_{i=w}^{s} \sum_{j=1}^{d} |\mathbf{A}_{i,p} \cdot \mathbf{V}_{p,j}|^2 = \sum_{i=w}^{s} |\mathbf{A}_{i,p}|^2 \cdot \sum_{j=1}^{d} |\mathbf{V}_{p,j}|^2 = \boxed{\sum_{i=w}^{s} |\mathbf{A}_{i,p}|^2 \cdot \|\mathbf{v}_p\|^2}. \tag{19}$$

This is also the value-pruning saliency score for evicting the $p$-th value vector $\mathbf{v}_p$ from the value cache. It is consistent with the expression in Equation 4 in Section 3.3.

### B.3 ISOLATED KEY-PRUNING SCORE

In isolated key pruning, the value-cache matrix is not perturbed and is assumed a constant. Therefore, minimizing the pruning-induced eviction error reduces to minimizing Equation 14. Same as in value-pruning score, we begin by deriving closed-form expressions for the gradient and Hessian with respect to $\widehat{\mathbf{K}}$. The first-order derivative of $\boldsymbol{E}_{i,j}$ with respect to $\widehat{\mathbf{K}}$ is:

$$\frac{\partial \boldsymbol{E}_{i,j}}{\partial \widehat{\mathbf{K}}} = \frac{\partial}{\partial \widehat{\mathbf{K}}} |\widehat{\mathbf{O}}_{i,j} - \mathbf{O}_{i,j}|^2 = 2(\widehat{\mathbf{O}}_{i,j} - \mathbf{O}_{i,j}) \frac{\partial \widehat{\mathbf{O}}_{i,j}}{\partial \widehat{\mathbf{K}}}. \tag{20}$$

Before deriving the first-order term explicitly, note that when evaluated at the point $(\mathbf{V}_{:,j}, \mathbf{K})$, we again have $\widehat{\mathbf{O}}_{i,j} - \mathbf{O}_{i,j} = 0$. Consequently, the first-order term is eliminated, a result that is the same as in isolated value pruning. We now explicitly derive the term $\widehat{\boldsymbol{M}}^{(i,j)} := \frac{\partial \widehat{\mathbf{O}}_{i,j}}{\partial \widehat{\mathbf{K}}}$ above:

$$\widehat{\boldsymbol{M}}_{p,r}^{(i,j)} = \left( \frac{\partial \widehat{\mathbf{O}}_{i,j}}{\partial \widehat{\mathbf{K}}} \right)_{p,r} = \frac{\partial}{\partial \widehat{\mathbf{K}}_{p,r}} \sum_{m=1}^{s} \widehat{\mathbf{A}}_{i,m} \cdot \widehat{\mathbf{V}}_{m,j} = \sum_{m=1}^{s} \frac{\partial \widehat{\mathbf{O}}_{i,j}}{\partial \widehat{\mathbf{A}}_{i,m}} \frac{\partial \widehat{\mathbf{A}}_{i,m}}{\partial \widehat{\mathbf{K}}_{p,r}}$$

$$= \sum_{m=1}^{s} \frac{\partial \widehat{\mathbf{O}}_{i,j}}{\partial \widehat{\mathbf{A}}_{i,m}} \frac{\partial}{\partial \widehat{\mathbf{K}}_{p,r}} \left( \frac{e^{\widehat{\mathbf{Z}}_{i,m}}}{\sum_{u=1}^{s} e^{\widehat{\mathbf{Z}}_{i,u}}} \right)$$

$$= \sum_{m=1}^{s} \frac{\partial \widehat{\mathbf{O}}_{i,j}}{\partial \widehat{\mathbf{A}}_{i,m}} \sum_{u=1}^{s} \frac{\widehat{\mathbf{A}}_{i,m}}{\widehat{\mathbf{Z}}_{i,u}} \frac{\widehat{\mathbf{Z}}_{i,u}}{\partial \widehat{\mathbf{K}}_{p,r}}$$

$$= \sum_{m=1}^{s} \widehat{\mathbf{V}}_{m,j} \sum_{u=1}^{s} \widehat{\mathbf{A}}_{i,m} (\delta_{mu} - \widehat{\mathbf{A}}_{i,u}) \frac{1}{\sqrt{d}} \mathbf{Q}_{i,r} \delta_{up}$$

$$= \sum_{m=1}^{s} \widehat{\mathbf{V}}_{m,j} \cdot \widehat{\mathbf{A}}_{i,m} (\delta_{mp} - \widehat{\mathbf{A}}_{i,p}) \frac{1}{\sqrt{d}} \mathbf{Q}_{i,r} \quad (\delta_{up} = 0 \text{ when } u \neq p)$$

$$= \frac{1}{\sqrt{d}} \mathbf{Q}_{i,r} \cdot \left( \sum_{m=1}^{s} \widehat{\mathbf{V}}_{m,j} \cdot \widehat{\mathbf{A}}_{i,m} \cdot \delta_{mp} - \sum_{m=1}^{s} \widehat{\mathbf{V}}_{m,j} \cdot \widehat{\mathbf{A}}_{i,m} \cdot \widehat{\mathbf{A}}_{i,p} \right)$$

$$= \frac{1}{\sqrt{d}} \mathbf{Q}_{i,r} \cdot \left( \widehat{\mathbf{V}}_{p,j} \cdot \widehat{\mathbf{A}}_{i,p} - \widehat{\mathbf{A}}_{i,p} \cdot \sum_{m=1}^{s} \widehat{\mathbf{V}}_{m,j} \cdot \widehat{\mathbf{A}}_{i,m} \right) \quad (\delta_{mp} = 0 \text{ when } m \neq p)$$

$$= \frac{1}{\sqrt{d}} \mathbf{Q}_{i,r} \cdot \widehat{\mathbf{A}}_{i,p} \cdot (\widehat{\mathbf{V}}_{p,j} - \widehat{\mathbf{O}}_{i,j}). \tag{21}$$

Therefore, the first-order derivative of $\boldsymbol{E}_{i,j}$ with respect to the perturbed key cache $\widehat{\mathbf{K}}$ is:

$$\frac{\partial \boldsymbol{E}_{i,j}}{\partial \widehat{\mathbf{K}}} = 2(\widehat{\mathbf{O}}_{i,j} - \mathbf{O}_{i,j})\widehat{\boldsymbol{M}}^{(i,j)}, \text{ where } \widehat{M}_{p,r}^{(i,j)} = \frac{1}{\sqrt{d}}\mathbf{Q}_{i,r} \cdot \widehat{\mathbf{A}}_{i,p} \cdot (\widehat{\mathbf{V}}_{p,j} - \widehat{\mathbf{O}}_{i,j}).$$

We now proceed to derive the Hessian, which is a 4-th order tensor of size $s \times d \times s \times d$. The second-order derivative of the element-wise perturbation $\boldsymbol{E}_{i,j}$ with respect to $\widehat{\mathbf{K}}$ is:

$$\frac{\partial^2 \boldsymbol{E}_{i,j}}{\partial \widehat{\mathbf{K}}^2} = \widehat{\boldsymbol{M}}^{(i,j)} \otimes \left( \frac{\partial}{\partial \widehat{\mathbf{K}}} 2(\widehat{\mathbf{O}}_{i,j} - \mathbf{O}_{i,j}) \right) + 2(\widehat{\mathbf{O}}_{i,j} - \mathbf{O}_{i,j})\frac{\partial \widehat{\boldsymbol{M}}^{(i,j)}}{\partial \widehat{\mathbf{K}}}$$

$$= 2\widehat{\boldsymbol{M}}^{(i,j)} \otimes \widehat{\boldsymbol{M}}^{(i,j)} + 2(\widehat{\mathbf{O}}_{i,j} - \mathbf{O}_{i,j})\frac{\partial \widehat{\boldsymbol{M}}^{(i,j)}}{\partial \widehat{\mathbf{K}}}, \tag{22}$$

where $\otimes$ denotes the matrix-wise outer product[1]. Since the Hessian will be evaluated at the point $(\mathbf{V}_{:,j}, \mathbf{K})$, where the perturbed attention output again matches the original, i.e., $\widehat{\mathbf{O}}_{i,j} - \mathbf{O}_{i,j} = 0$, the second term in Equation 22 vanishes. Therefore, the Hessian is:

$$\frac{\partial^2 \boldsymbol{E}_{i,j}}{\partial \widehat{\mathbf{K}}^2}\bigg|_{(\mathbf{V}_{:,j}, \mathbf{K})} = 2\boldsymbol{M}^{(i,j)} \otimes \boldsymbol{M}^{(i,j)} = 2\mathrm{vec}(\boldsymbol{M}^{(i,j)})^{\top}\mathrm{vec}(\boldsymbol{M}^{(i,j)}), \tag{23}$$

$$\text{where } M_{p,r}^{(i,j)} = \frac{1}{\sqrt{d}}\mathbf{Q}_{i,r} \cdot \mathbf{A}_{i,p} \cdot (\mathbf{V}_{p,j} - \mathbf{O}_{i,j}).$$

Here, we use $\mathrm{vec}(\cdot)$ to denote the vectorization of a matrix. Substituting the gradient and Hessian back into Equation 14, we get the pruning-induced eviction error when key states are considered as isolate pruning units:

$$\mathcal{L}^{\mathrm{key}} = \sum_{i=w}^{s} \sum_{j=1}^{d} \mathrm{vec}(\delta\mathbf{K})^{\top}\mathrm{vec}(\boldsymbol{M}^{(i,j)})^{\top}\mathrm{vec}(\boldsymbol{M}^{(i,j)})\mathrm{vec}(\delta\mathbf{K})$$

$$= \sum_{i=w}^{s} \sum_{j=1}^{d} |\mathrm{vec}(\boldsymbol{M}^{(i,j)})\mathrm{vec}(\delta\mathbf{K})|^2. \tag{24}$$

Similarly, the row-wise pruning constraint in $e_p^{\top}\widehat{\mathbf{K}} = \mathbf{0}$ implies that when pruning the $p$-th token position, the key cache perturbation $\delta\mathbf{K}$ is explicitly in the form:

$$\delta\mathbf{K}_{t,j} = \begin{cases} -\mathbf{K}_{p,j}, & \text{when } t = p \\ 0, & \text{when } t \neq p \end{cases}, \quad \forall j = 1, ..., d.$$

Substituting this key perturbation into $\mathcal{L}^{\mathrm{key}}$, we obtain the output perturbation when pruning the $p$-th value vector from the value cache $\mathbf{K}$ (we let $\mathbf{m}_p^{(i,j)}$ to denote the $p$-th row vector of $\boldsymbol{M}^{(i,j)}$):

$$\boldsymbol{S}_p^{\mathrm{key}} = \sum_{i=w}^{s} \sum_{j=1}^{d} |\mathbf{m}_p^{(i,j)} \mathbf{k}_p|^2 = \sum_{i=w}^{s} \sum_{j=1}^{d} |\sum_{r=1}^{d} \frac{1}{\sqrt{d}}\mathbf{Q}_{i,r} \cdot \mathbf{A}_{i,p}(\mathbf{V}_{p,j} - \mathbf{O}_{i,j}) \cdot \mathbf{K}_{p,r}|^2$$

$$= \frac{1}{d} \sum_{i=w}^{s} \sum_{j=1}^{d} |\mathbf{A}_{i,p}(\mathbf{V}_{p,j} - \mathbf{O}_{i,j}) \cdot \sum_{r=1}^{d} \mathbf{Q}_{i,r}\mathbf{K}_{p,r}|^2$$

$$= \frac{1}{d} \sum_{i=w}^{s} \sum_{j=1}^{d} |\mathbf{A}_{i,p}(\mathbf{V}_{p,j} - \mathbf{O}_{i,j}) \cdot \mathbf{q}_i\mathbf{k}_p^{\top}|^2$$

$$= \frac{1}{d} \sum_{i=w}^{s} \left( |\mathbf{A}_{i,p}|^2 \cdot |\mathbf{q}_i\mathbf{k}_p^{\top}|^2 \sum_{j=1}^{d} |\mathbf{V}_{p,j} - \mathbf{O}_{i,j}|^2 \right)$$

$$\boxed{= \sum_{i=w}^{s} |\mathbf{A}_{i,p}|^2 \cdot |\mathbf{Z}_{i,p}|^2 \cdot \|\mathbf{v}_p - \mathbf{o}_i\|_2^2}.$$

This is also the key-pruning saliency score for evicting the $p$-th key vector $\mathbf{k}_p$ from the key cache. It is consistent with the expression in Equation 5 in Section 3.3.

---

[1]For any two matrices $\boldsymbol{C}, \boldsymbol{D} \in \mathbb{R}^{s \times d}$, the output tensor via outer product is $\boldsymbol{C} \otimes \boldsymbol{D} \in \mathbb{R}^{s \times d \times s \times d}$.

### B.4 JOINT-PRUNING SCORE

When both $\mathbf{V}_{:,j}$ and $\mathbf{K}$ are treated as variables affecting $\mathcal{L}$, we need to compute the cross-term in the error function, which corresponds to deriving a closed-form expression for Equation 15. Given that the first-order derivative of $\boldsymbol{E}_{i,j}$ with respect to $\widehat{\mathbf{V}}_{:,j}$ is known, we can compute the cross-term as follows:

$$\frac{\partial^2 \boldsymbol{E}_{i,j}}{\partial \widehat{\mathbf{V}}_{:,j} \partial \widehat{\boldsymbol{K}}} = \frac{\partial}{\partial \widehat{\boldsymbol{K}}} 2(\widehat{\mathbf{O}}_{i,j} - \mathbf{O}_{i,j}) \hat{\mathbf{a}}_i$$

$$= \hat{\mathbf{a}}_i \otimes \left( \frac{\partial}{\partial \widehat{\boldsymbol{K}}} 2(\widehat{\mathbf{O}}_{i,j} - \mathbf{O}_{i,j}) \right) + 2(\widehat{\mathbf{O}}_{i,j} - \mathbf{O}_{i,j}) \frac{\partial \hat{\mathbf{a}}_i}{\partial \widehat{\boldsymbol{K}}}. \tag{25}$$

Same as in deriving key-pruning scores, the second term above will be zero when evaluated at the point $(\mathbf{V}_{:,j}, \mathbf{K})$. Therefore, the cross second-order term is:

$$\frac{\partial^2 \boldsymbol{E}_{i,j}}{\partial \widehat{\mathbf{V}}_{:,j} \partial \widehat{\boldsymbol{K}}} \bigg|_{(\mathbf{V}_{:,j}, \mathbf{K})} = 2\mathbf{a}_i \otimes \boldsymbol{M}^{(i,j)}, \text{ where } M_{p,r}^{(i,j)} = \frac{1}{\sqrt{d}} \mathbf{Q}_{i,r} \cdot \mathbf{A}_{i,p} \cdot (\mathbf{V}_{p,j} - \mathbf{O}_{i,j}). \tag{26}$$

Substituting this back into Equation 15, we get the cross-term of the pruning-induced eviction error when the the value and key states are considered as combined pruning units:

$$\mathcal{L}^{\text{cross}} = \sum_{i=w}^{s} \sum_{j=1}^{d} \delta \mathbf{V}_{:,j}^{\top} \big( 2\mathbf{a}_i \otimes \boldsymbol{M}^{(i,j)} \big) \delta \mathbf{K} = 2 \sum_{i=w}^{s} \sum_{j=1}^{d} (\delta \mathbf{V}_{:,j}^{\top} \mathbf{a}_i) \cdot \langle \boldsymbol{M}^{(i,j)}, \delta \mathbf{K} \rangle_F, \tag{27}$$

where $\langle \boldsymbol{M}^{(i,j)}, \delta \mathbf{K} \rangle_F = \sum_{p,r} M_{p,r}^{(i,j)} \cdot \delta \mathbf{K}_{p,r} = \mathbf{tr}(\boldsymbol{M}^{(i,j)\top} \delta \mathbf{K})$ is the Frobenius inner product.

When the $p$-th key and value are pruned, the row-wise pruning constraint in $\boldsymbol{e}_p^{\top} [\widehat{\mathbf{V}} \ \widehat{\mathbf{K}}] = \mathbf{0}$ implies that the value cache perturbation $\delta \mathbf{V}$ and the key cache perturbation $\delta \mathbf{K}$ are explicitly in the form:

$$\delta \mathbf{V}_{t,j} = \begin{cases} -\mathbf{V}_{p,j}, & \text{when } t = p \\ 0, & \text{when } t \neq p \end{cases}, \quad \delta \mathbf{K}_{t,j} = \begin{cases} -\mathbf{K}_{p,j}, & \text{when } t = p \\ 0, & \text{when } t \neq p \end{cases}, \quad \forall \, j = 1, ..., d.$$

Thus, we have each term in $\mathcal{L}^{\text{cross}}$:

$$\delta \mathbf{V}_{:,j}^{\top} \mathbf{a}_i = -\mathbf{A}_{i,p} \mathbf{V}_{p,j}, \quad \langle \boldsymbol{M}^{(i,j)}, \delta \mathbf{K} \rangle_F = -\frac{1}{\sqrt{d}} \mathbf{A}_{i,p} \cdot (\mathbf{V}_{p,j} - \mathbf{O}_{i,j}) \cdot (\mathbf{q}_i \mathbf{k}_p^{\top}).$$

Substituting these two expressions back into Equation 27, we obtain the cross term in closed form:

$$\mathcal{L}^{\text{cross}} = \frac{2}{\sqrt{d}} \sum_{i=w}^{s} \sum_{j=1}^{d} \mathbf{A}_{i,p}^2 \cdot \mathbf{V}_{p,j} \cdot (\mathbf{V}_{p,j} - \mathbf{O}_{i,j}) \cdot (\mathbf{q}_i \mathbf{k}_p^{\top})$$

$$= \frac{2}{\sqrt{d}} \sum_{i=w}^{s} \mathbf{A}_{i,p}^2 \cdot (\mathbf{q}_i \mathbf{k}_p^{\top}) \sum_{j=1}^{d} \mathbf{V}_{p,j} (\mathbf{V}_{p,j} - \mathbf{O}_{i,j})$$

$$= \boxed{2 \sum_{i=w}^{s} \mathbf{A}_{i,p}^2 \cdot \mathbf{Z}_{i,p} \cdot (\|\mathbf{v}_p\|_2^2 - \mathbf{v}_p^{\top} \mathbf{o}_i)}. \tag{28}$$

Finally, the saliency score of the token at position $p$ when both value and key states are treated as pruning units is the summation of $\mathcal{L}^{\text{value}}$, $\mathcal{L}^{\text{key}}$ and $\mathcal{L}^{\text{cross}}$:

$$\boxed{\boldsymbol{S}_p^{\text{joint}} = 2 \sum_{i=w}^{s} \mathbf{A}_{i,p}^2 \cdot \mathbf{Z}_{i,p} \cdot (\|\mathbf{v}_p\|_2^2 - \mathbf{v}_p^{\top} \mathbf{o}_i) + \boldsymbol{S}_p^{\text{value}} + \boldsymbol{S}_p^{\text{key}}}. \tag{29}$$

This is consistent with the joint score expression in Equation 6 of Section 3.3.

## B.5 Scores for Grouped-Query Attention

In models with Grouped-Query Attention (GQA) (Ainslie et al., 2023), multiple query heads share a single key and value head, reducing KV cache storage at the architectural level. To support cache eviction under this setting, we explicitly incorporate the head-group structure into our analysis and derive the corresponding OBCACHE scores.

We use superscripts $g$ to denote KV-head index, $h$ to denote query-head index, and $H(g)$ as the set of query heads associated with KV head $g$. Our previous derivation omits head indices for simplicity. Here, we specifically include head indexing for multi-head attention (MHA). If the pruning-induced eviction error is defined on the per-head attention output,

$$\mathcal{L} := \|\widehat{\mathbf{O}}^h - \mathbf{O}^h\|_F^2, \text{ where } \mathbf{O}^h, \widehat{\mathbf{O}}^h \in \mathbb{R}^{s \times d},$$

the OBCACHE scores for the $h$-th head naturally extend to:

$$\mathbf{S}_{p,h}^{\text{value}} = \sum_i |\mathbf{A}_{i,p}^h|^2 \cdot \|\mathbf{v}_p^h\|^2 \tag{30}$$

$$\mathbf{S}_{p,h}^{\text{key}} = \sum_i |\mathbf{A}_{i,p}^h|^2 \cdot |\mathbf{Z}_{i,p}^h|^2 \cdot \|\mathbf{v}_p^h - \mathbf{o}_i^h\|_2^2 \tag{31}$$

$$\mathbf{S}_{p,h}^{\text{joint}} = 2 \sum_i |\mathbf{A}_{i,p}^h|^2 \cdot \mathbf{Z}_{i,p}^h \cdot (\|\mathbf{v}_p^h\|_2^2 - \mathbf{v}_p^{h\top}\mathbf{o}_i^h) + \mathbf{S}_{p,h}^{\text{value}} + \mathbf{S}_{p,h}^{\text{key}} \tag{32}$$

In standard MHA, each output head $\mathbf{O}^h$ depends on its own $\mathbf{K}^h$ and $\mathbf{V}^h$. In GQA, however, a group of output heads $\{\mathbf{O}^h\}_{h \in H(g)}$ shares the same key and value $\mathbf{K}^g$ and $\mathbf{V}^g$, i.e.,

$$\mathbf{V}^g \equiv \mathbf{V}^h, \forall h \in H(g), \quad \mathbf{K}^g \equiv \mathbf{K}^h, \forall h \in H(g)$$

If we now modify the objective to be the sum of output perturbation across heads within a group:

$$\mathcal{L} := \sum_{h \in H(g)} \|\widehat{\mathbf{O}}^h - \mathbf{O}^h\|_F^2,$$

the OBCACHE scores for the $g$-th KV head introduce an additional summation over its associated query heads:

$$\mathbf{S}_{p,g}^{\text{value}} = \sum_{h \in H(g)} \mathbf{S}_{p,h}^{\text{value}} = \sum_{h \in H(g)} \sum_i |\mathbf{A}_{i,p}^h|^2 \cdot \|\mathbf{v}_p^g\|^2 \tag{33}$$

$$\mathbf{S}_{p,g}^{\text{key}} = \sum_{h \in H(g)} \mathbf{S}_{p,h}^{\text{key}} = \sum_{h \in H(g)} \sum_i |\mathbf{A}_{i,p}^h|^2 \cdot |\mathbf{Z}_{i,p}^h|^2 \cdot \|\mathbf{v}_p^g - {}_i^h\|_2^2 \tag{34}$$

$$\mathbf{S}_{p,g}^{\text{joint}} = \sum_{h \in H(g)} \mathbf{S}_{p,h}^{\text{joint}} = 2 \sum_{h \in H(g)} \left( \sum_i |\mathbf{A}_{i,p}^h|^2 \cdot \mathbf{Z}_{i,p}^h \cdot (\|\mathbf{v}_p^g\|_2^2 - \mathbf{v}_p^{g\top}\mathbf{o}_i^h) \right) + \mathbf{S}_{p,g}^{\text{value}} + \mathbf{S}_{p,g}^{\text{key}} \tag{35}$$

Objective formulations other than summation can also yield valid per-KV-head scores. In our experiments, we follow the SnapKV implementation, which stores the KV cache for all query heads and performs eviction across them. We additionally evaluate the above GQA-aware formulation without retaining all query heads, and the results are reported in Table 3 in Appendix D.2.

## C  EXPERIMENTAL SETUP

All the experiments are conducted on a single NVIDIA A100 GPU. We adapt the Hugging Face Transformers library (Wolf et al., 2020) with PyTorch (Paszke et al., 2019) to implement the cache eviction algorithms. We use two representative instruction-tuned large language models as backbones: LLaMA-3.1-8B-Instruct (Grattafiori et al., 2024) and Qwen-2.5-7B-Instruct (Bai et al., 2023a). Both models employ Grouped Query Attention (GQA) and natively support a context window size of up to 128K tokens. All model inference is performed in half precision (bfloat16) with no quantization applied.

For tasks with extensive prefill prompt lengths, we use FlashAttention-2 (Dao, 2023) for prefill attention computation to reduce memory overhead. We then recompute attention weights for selected query positions as needed to compute saliency scores for token eviction, since attention weights are not materialized in FlashAttention.

### C.1  DATASETS

We evaluate OBCACHE and existing cache eviction methods on three benchmarks: Needle-In-A-Haystack (Kamradt, 2023), LongBench (Bai et al., 2023b), and perplexity on the PG19 dataset (Rae et al., 2019), targeting prefill-phase static cache eviction and decoding-phase dynamic cache eviction, respectively.

**Needle-In-A-Haystack (NIAH).**  We follow the setup from the RULER benchmark (Hsieh et al., 2024), specifically the `niah_single_2` task[2], where a randomly generated 7-digit "needle" passkey is embedded into a Paul Graham essay haystack. Context lengths of 4K, 8K, 16K, and 32K tokens are tested, with each setting containing 250 randomly generated samples.

**LongBench.**  LongBench includes 16 datasets across six task categories: single-document QA, multi-document QA, summarization, few-shot learning, synthetic reasoning, and code completion. The average input length is 6,711 words (approximately 16K tokens). For all models and methods, we truncate each sample to a maximum of 32K tokens. To ensure fair comparison, we follow SnapKV (Li et al., 2024) and perform cache eviction only during the prefill phase. Task-specific evaluation metrics (e.g., Exact Match/F1 for QA tasks, ROUGE for summarization tasks) are reported using LongBench's official evaluation script[3].

**Perplexity.**  For decoding-phase cache eviction, we adopt the PG19 test set following the setup in StreamingLLM (Xiao et al., 2024). PG19 contains 100 full-length books, each averaging around 70K tokens. We evaluate on the standard test split[4] and compute perplexity across varying context lengths (from 1 to 32K), using a fixed KV cache budget of 1024 tokens for all dynamic cache eviction methods.

### C.2  BASELINE SETUPS

#### C.2.1  PREFILL EVICTION

To evaluate baselines for prefill-phase cache eviction, we follow the experimental setup of SnapKV (Li et al., 2024) and perform cache eviction only during the prefill phase. Since SnapKV does not support eviction during decoding, no cached tokens are evicted in the decoding phase (for NIAH and LongBench tasks). In these benchmarks, the prompt length dominates and is the primary bottleneck; thus, applying decoding-phase eviction has little impact on performance.

**$H_2O$.**  $H_2O$ originally accumulates attention weights across all historical query positions to make eviction decisions. This is incompatible with the FlashAttention prefill implementation, as it requires recomputing full attention weights. To enable efficient implementation, we follow SnapKV's version of $H_2O$ and accumulate query positions only within a recent perturbation window. For NIAH tasks,

---

[2] https://github.com/NVIDIA/RULER/blob/main/README.md

[3] https://github.com/THUDM/LongBench/blob/main/LongBench/pred.py

[4] https://huggingface.co/datasets/emozilla/pg19

Table 2: Comparison of OBCACHE scores with purely attention-based scores on cache eviction. We define $W = s - w$ as the size of the perturbation window, $d_{\text{head}}$ as the head hidden dimension.

| Method | Saliency Score $\boldsymbol{S}_p$ | Complexity |
|---|:---:|:---:|
| Attention-Based | $\sum_i |\mathbf{A}_{i,p}|$ | $O(W)$ |
| OBCACHE-VALUE | $\sum_i |\mathbf{A}_{i,p}|^2 \cdot \|\mathbf{v}_p\|^2$ | $O(W + d_{\text{head}})$ |
| OBCACHE-KEY | $\sum_i |\mathbf{A}_{i,p}|^2 \cdot |\mathbf{Z}_{i,p}|^2 \cdot \|\mathbf{v}_p - \mathbf{o}_i\|_2^2$ | $O(W d_{\text{head}})$ |
| OBCACHE-JOINT | $2\sum_i |\mathbf{A}_{i,p}|^2 \cdot \mathbf{Z}_{i,p} \cdot (\|\mathbf{v}_p\|_2^2 - \mathbf{v}_p^\top \mathbf{o}_i) + \boldsymbol{S}_p^{\text{value}} + \boldsymbol{S}_p^{\text{key}}$ | $O(W d_{\text{head}})$ |

the perturbation window size is set to 16. For LongBench tasks, the window size is $5\%$ of the prompt length $l$. All tokens in the perturbation window are treated as recent tokens, and the remaining cache budget is allocated to select heavy hitters.

**TOVA.** TOVA does not accumulate attention weights but makes eviction decisions solely based on the most recent attention distribution. In all prefill-eviction evaluations, the cache budget is entirely allocated to the heavy hitters.

**SnapKV.** In SnapKV, the same perturbation window as in $H_2O$ is used. The key difference is that SnapKV additionally applies a max-pooling filter to the accumulated scores before eviction. For all SnapKV and OBCACHE-enhanced experiments, we follow the official implementation, using a max-pooling function with a kernel size of 7 and a stride of 1.

### C.2.2 DECODING EVICTION

To evaluate baselines for decoding-phase cache eviction, we follow the setup of StreamingLLM (Xiao et al., 2024), where eviction decisions are made at every decoding step. All eviction methods are evaluated with a fixed 1024-token cache budget.

**StreamingLLM.** For StreamingLLM, we maintain the first 4 tokens as attention sinks and always keep the most recent 1020 tokens. Since attention sinks are essential for long-context generation, we retain the first 4 sink tokens in all other methods as well.

**$H_2O$.** At each decoding step, $H_2O$ accumulates attention weights across all historical query positions to make eviction decisions. In all perplexity experiments of $H_2O$ and its OBCACHE-enhanced variants, a 256-token recent window is always reserved. The remaining 764 tokens are dynamically selected using their respective saliency scores.

**TOVA.** As in prefill eviction, TOVA does not use a fixed recent window. All 1020 heavy-hitter tokens are dynamically selected based on the attention scores from the latest query position.

## D EXPERIMENTAL RESULTS

### D.1 EFFICIENCY EVALUATION OF OBCACHE SCORES

To analyze the additional computation overhead introduced by OBCACHE scores, we compare their complexity against purely attention-based scores, as summarized in Table 2. For OBCACHE-VALUE, the only additional operation beyond attention-based scores is computing the norm of the value state, which adds a negligible linear term in the per-head hidden dimension. OBCACHE-KEY and OBCACHE-JOINT require computing the norm of the difference between the value vector and each attention output vector within the perturbation window, resulting in a complexity of $O(W d_{\text{head}})$. This is more expensive than OBCACHE-VALUE. However, in practical settings, the perturbation window $W$ is often very small (e.g., $W = 16$ in the prefill phase and $W = 1$ during decoding), so the additional overhead remains minor relative to the overall model computation.

We empirically benchmark peak memory usage and latency overhead during prefill and decoding with varying context lengths and batch sizes on a single A100-80GB GPU, as shown in Figure 5

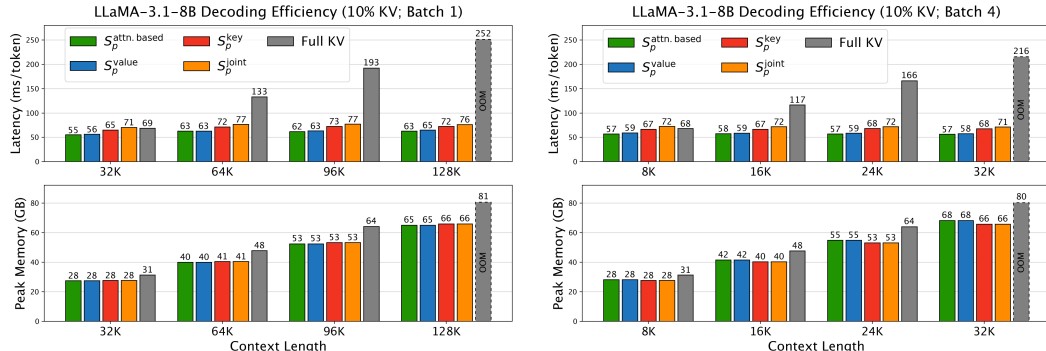

Figure 5: Decoding complexity of OBCACHE. We report the per-token decoding latency (averaged over 512 generated tokens) and peak memory consumption across different methods. Results are shown for varying context lengths, with batch size 1 on the left and batch size 4 on the right. Out-of-memory (OOM) results are linearly extrapolated from measured data.

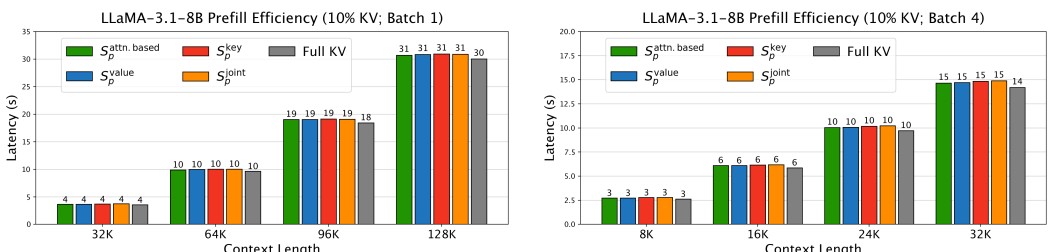

Figure 6: Prefill complexity of OBCACHE. We report the time required from the start of prefill until the first token is generated, including the cache eviction operation. Results are presented for different context lengths, with batch size 1 shown on the left and batch size 4 on the right.

and Figure 6. For each configuration, we measure: (1) the average time from the start of prefill until the first output token is generated, including the cache eviction operation, and (2) the per-token decoding latency where scoring and eviction are dynamically updated.

Across both single- and multi-batch settings, the results show that OBCACHE-VALUE has nearly identical decoding latency ($< 2$ ms) to attention-based methods across all context lengths. While OBCACHE-KEY and OBCACHE-JOINT introduce additional latency ($< 15$ ms), their cost remains substantially lower than full-cache decoding and does not scale linearly with context length, demonstrating practical efficiency. In the prefill phase, all eviction methods show negligible additional latency compared to the full-KV baseline. These empirical results align with the complexity analysis and demonstrate that OBCache scores can be efficiently computed. We also note that our implementation of score-based cache eviction is built on default PyTorch primitives (e.g., `torch.gather` and `torch.cat`) without customized kernels. As a result, at small context lengths, eviction methods exhibit higher latency than full-cache decoding. In the future, developing specialized kernels for score computation and cache eviction would yield additional speed improvements and further enhance practical deployment efficiency.

## D.2 RESULTS USING GQA SCORES

We evaluate the GQA-aware OBCACHE scores in Appendix B.5 on Needle-In-A-Haystack, three single-document QA tasks (Qasper, MultifieldQA, and NarrativeQA), one summarization task (MultiNews), and one few-shot learning task (SAMSum) from LongBench. The results, compared against those obtained when storing all query heads, are presented in Table 3. We find that adopting GQA-aware scores can substantially improve performance in retrieval-centric tasks (e.g., NIAH and Qasper) while requiring fewer KV heads. However, on other tasks, the GQA setting leads to similar or inferior performance.

Table 3: Results of NIAH, Qasper, MultifieldQA, NarrativeQA, MultiNews and SAMSum on LLaMA-3.1-8B-Instruct using GQA-aware scores. In NIAH, we use 8K context lengths and set the KV budget at 80 tokens. In other LongBench tasks, we use 10% cache budget with other settings unchanged.

| | NIAH (8K) | | Qasper | | MultifieldQA | | NarrativeQA | | MultiNews | | SAMSum | |
|---|---|---|---|---|---|---|---|---|---|---|---|---|
| | w/o | w/ GQA score | w/o | w/ GQA score | w/o | w/ GQA score | w/o | w/ GQA score | w/o | w/ GQA score | w/o | w/ GQA score |
| All KV | 100.0 | | 44.75 | | 55.21 | | 28.79 | | 27.23 | | 43.75 | |
| H2O | 10.8 | 68.0 | 31.95 | 35.29 | 49.83 | 51.97 | 26.94 | **29.39** | 22.05 | 21.45 | 42.3 | 41.63 |
| + OBCACHE-VALUE | 21.6 | **80.0** | 32.65 | **37.46** | 50.55 | **53.01** | 27.55 | 29.26 | 21.67 | 22.21 | 42.24 | 42.31 |
| + OBCACHE-KEY | 29.6 | 72.0 | 35.55 | 36.09 | 50.82 | 52.33 | 27.65 | 29.2 | 22.2 | 22.27 | 42.18 | 42.5 |
| + OBCACHE-JOINT | 33.2 | 74.8 | 34.29 | 36.52 | 52.17 | 52.32 | 27.53 | 29.17 | 22.19 | **22.39** | **43.07** | 42.28 |
| TOVA | 9.2 | 31.6 | 28.51 | 30.37 | 48.16 | 51.78 | 27.11 | 26.71 | 21.41 | 21.31 | 42.34 | 43.91 |
| + OBCACHE-VALUE | 10.4 | 52.4 | 29.23 | **32.01** | 47.19 | **52.33** | 26.35 | **27.25** | 21.46 | **21.91** | 42.82 | 43.77 |
| + OBCACHE-KEY | 12.4 | **55.6** | 29.93 | 30.66 | 49.25 | 51.57 | 26.94 | 27.23 | 21.51 | 21.65 | 42.44 | **44.06** |
| + OBCACHE-JOINT | 12.8 | **55.6** | 29.78 | 30.98 | 49.22 | 51.44 | 26.18 | 27.07 | 21.5 | 21.67 | 42.79 | 43.34 |
| SnapKV | 58.4 | 98.4 | 37.95 | 38.31 | 53.65 | 53.96 | 28.09 | 28.68 | 22.2 | 21.86 | 42.27 | 41.78 |
| + OBCACHE-VALUE | 63.6 | **99.2** | 37.77 | **40.56** | 53.89 | **54.4** | 28.38 | 27.83 | 22.57 | 22.34 | **43.5** | 43.26 |
| + OBCACHE-KEY | 72.8 | 98.8 | 39.71 | 39.48 | 53.44 | 53.3 | 28.22 | **29.11** | **22.73** | 22.37 | 42.58 | 42.74 |
| + OBCACHE-JOINT | 72.0 | 97.2 | 38.2 | 40.5 | 53.92 | 53.71 | 28.69 | 28.12 | 22.6 | 21.97 | 43.16 | 42.89 |

Table 4: LongBench results for LLaMA-3.1-8B-Instruct.

| | Methods | Single-Document QA | | | Multi-Document QA | | | Summarization | | | Few-shot Learning | | | Synthetic | | Code | |
|---|---|---|---|---|---|---|---|---|---|---|---|---|---|---|---|---|---|
| | | NrtvQA | Qasper | MF-en | HotpotQA | 2WikiMQA | Musique | GovReport | QMSum | MultiNews | TREC | TriviaQA | SAMSum | PCount | PRe | Lcc | RB-P |
| | ALL KV | 28.79 | 44.75 | 55.21 | 55.53 | 45.59 | 30.89 | 35.01 | 25.39 | 27.23 | 72.5 | 91.65 | 43.75 | 6.78 | 99.5 | 52.23 | 49.39 |
| 5% KV | H2O | 24.81 | 30.09 | 47.46 | 52.38 | **42.26** | 28.35 | 25.92 | 23.71 | 20.87 | 40.5 | 90.81 | 41.32 | 6.15 | 99.5 | 40.46 | 44.56 |
| | + OBCACHE-VALUE | 26.26 | 29.7 | 48.15 | 52.43 | 41.66 | 28.27 | 26.29 | **24.06** | 21.03 | 40.5 | 91.58 | **41.72** | 6.31 | 99.5 | 41.16 | 44.63 |
| | + OBCACHE-KEY | 25.43 | **32.48** | 47.27 | 52.91 | 41.66 | **28.94** | 25.61 | 23.93 | 21.17 | 41.5 | 91.13 | 41.69 | 6.31 | 99.5 | **41.6** | **45.24** |
| | + OBCACHE-JOINT | 25.61 | 32.2 | **49.47** | **53.31** | 41.58 | 28.22 | **26.34** | 23.69 | **21.28** | 41.0 | 90.78 | 41.53 | 6.31 | 99.5 | 41.26 | 45.16 |
| | TOVA | 26.27 | 24.83 | 44.48 | 52.69 | 39.04 | 28.68 | 25.0 | 22.96 | 20.71 | 41.0 | 92.16 | 42.28 | 6.31 | 99.5 | 37.35 | 35.21 |
| | + OBCACHE-VALUE | 26.41 | 25.76 | 45.03 | 53.0 | 39.61 | 28.5 | 25.12 | 22.93 | **20.77** | 42.0 | **92.66** | **42.49** | 6.31 | 99.5 | 37.33 | 35.56 |
| | + OBCACHE-KEY | **26.71** | 25.72 | **45.89** | 53.3 | **40.61** | **28.85** | 25.39 | 22.55 | 20.35 | **43.0** | 91.68 | 41.96 | 6.31 | 99.5 | 37.02 | 34.29 |
| | + OBCACHE-JOINT | 25.93 | **25.88** | 43.6 | 53.03 | 39.89 | 28.13 | 25.23 | **23.09** | **20.77** | 42.0 | 91.61 | 42.24 | 6.31 | 99.5 | **38.15** | **36.3** |
| | SnapKV | 26.34 | 35.2 | **54.33** | 54.41 | 42.96 | 30.06 | 25.84 | 24.06 | 20.28 | 67.5 | **91.72** | 41.64 | 6.31 | 99.5 | 42.82 | 46.58 |
| | + OBCACHE-VALUE | 26.59 | 35.34 | 51.69 | 55.03 | 42.3 | 30.03 | 26.05 | **24.73** | 20.25 | 66.0 | 91.68 | **41.66** | 6.31 | 99.5 | 42.82 | 46.47 |
| | + OBCACHE-KEY | **27.01** | **36.25** | 52.22 | **55.2** | 42.86 | 30.39 | **26.3** | 24.42 | 20.47 | **68.0** | 90.72 | 41.35 | 6.31 | 99.5 | 42.57 | **47.5** |
| | + OBCACHE-JOINT | 26.8 | 35.22 | 53.27 | 54.74 | **43.33** | **30.47** | 26.13 | 24.48 | **20.56** | 67.5 | 91.34 | 41.34 | 6.31 | 99.5 | **42.9** | 47.07 |
| 10% KV | H2O | 26.94 | 31.95 | 49.83 | 53.37 | 44.41 | 29.22 | 27.4 | **23.86** | 22.05 | 43.5 | 91.51 | 42.3 | 6.31 | 99.5 | 47.35 | 46.99 |
| | + OBCACHE-VALUE | 27.55 | 32.65 | 50.55 | **53.99** | **45.25** | **30.2** | 27.33 | 23.61 | 21.67 | 43.5 | **91.83** | 42.24 | 6.31 | 99.5 | 48.02 | 46.59 |
| | + OBCACHE-KEY | **27.65** | **35.55** | 50.82 | 53.39 | 44.53 | 29.66 | 27.26 | 23.72 | **22.2** | **45.0** | 91.52 | 42.18 | 6.31 | 99.5 | **48.25** | **47.03** |
| | + OBCACHE-JOINT | 27.53 | 34.29 | **52.17** | 53.65 | 44.71 | 29.63 | **27.76** | 23.84 | 22.19 | 44.0 | 91.51 | **43.07** | 6.31 | 99.5 | 48.21 | 47.01 |
| | TOVA | **27.11** | 28.51 | 48.16 | 54.13 | 39.84 | 28.24 | 26.66 | 23.0 | 21.41 | 46.0 | **92.13** | 42.34 | 6.31 | 99.5 | 41.32 | 38.36 |
| | + OBCACHE-VALUE | 26.35 | 29.23 | 47.19 | **54.58** | 40.09 | 28.92 | 26.56 | 23.22 | 21.46 | 46.5 | 91.68 | **42.82** | 6.31 | 99.5 | 41.21 | 38.99 |
| | + OBCACHE-KEY | 26.94 | **29.93** | **49.25** | 54.31 | 40.3 | 28.87 | **26.73** | 23.18 | **21.51** | **47.5** | 91.78 | 42.44 | 6.31 | 99.5 | 41.04 | **39.36** |
| | + OBCACHE-JOINT | 26.18 | 29.78 | 49.22 | 54.3 | **40.73** | **29.19** | 26.47 | **23.71** | 21.5 | 47.0 | 91.47 | 42.79 | 6.31 | 99.5 | **41.57** | 39.17 |
| | SnapKV | 28.09 | 37.95 | 53.65 | 55.08 | 43.53 | 30.17 | 27.79 | **24.41** | 22.2 | 67.5 | **91.97** | 42.27 | 6.31 | 99.5 | 50.07 | 48.09 |
| | + OBCACHE-VALUE | 28.38 | 37.77 | 53.89 | **55.67** | 43.0 | 30.05 | 27.7 | 23.96 | 22.57 | 68.0 | **91.97** | **43.5** | 6.31 | 99.5 | 49.99 | 48.48 |
| | + OBCACHE-KEY | 28.22 | **39.71** | 53.44 | 54.98 | **44.12** | 30.24 | **27.96** | 24.37 | **22.73** | 68.5 | 91.18 | 42.58 | 6.31 | 99.5 | **50.18** | **48.61** |
| | + OBCACHE-JOINT | **28.69** | 38.2 | **53.92** | 55.33 | 44.1 | **30.35** | 27.63 | 24.38 | 22.6 | 67.5 | 91.75 | 43.16 | 6.31 | 99.5 | 49.46 | 48.5 |
| 20% KV | H2O | **28.9** | 35.96 | 51.73 | **55.63** | 44.3 | 29.69 | **29.39** | 24.04 | 23.03 | 47.5 | 91.27 | 42.31 | 6.31 | 99.5 | 50.1 | **48.46** |
| | + OBCACHE-VALUE | 28.78 | 37.19 | 52.51 | 55.0 | **45.19** | **30.49** | 29.34 | 24.32 | 23.3 | 48.0 | **91.77** | 42.71 | 6.56 | 99.5 | 51.43 | 47.9 |
| | + OBCACHE-KEY | 28.36 | 37.41 | 52.65 | 54.98 | 44.82 | 29.87 | 29.17 | **24.38** | 23.31 | **50.0** | 91.74 | **42.8** | 6.31 | 99.5 | 51.34 | 48.38 |
| | + OBCACHE-JOINT | 28.26 | **38.42** | **53.47** | 55.38 | 44.92 | 29.99 | 29.11 | 24.23 | 23.09 | 48.5 | 91.44 | 42.66 | 6.31 | 99.5 | **51.45** | 48.22 |
| | TOVA | 27.47 | 33.83 | **52.88** | 53.79 | 41.74 | 29.57 | 28.94 | 23.61 | 22.71 | 62.0 | 91.59 | 42.2 | 6.31 | 99.5 | 44.34 | 41.7 |
| | + OBCACHE-VALUE | 27.15 | 34.42 | 52.38 | **54.54** | 42.22 | **30.47** | 28.92 | 23.59 | 23.02 | **62.5** | 91.39 | **42.67** | 6.31 | 99.5 | 44.41 | 42.7 |
| | + OBCACHE-KEY | **27.52** | **35.09** | 52.09 | 53.8 | **42.42** | 30.14 | 28.85 | **23.63** | **23.06** | 62.0 | 91.39 | 42.4 | 6.31 | 99.5 | **45.36** | **43.05** |
| | + OBCACHE-JOINT | 27.11 | 34.97 | 52.29 | 54.35 | 42.39 | 30.14 | **29.03** | 23.56 | 23.05 | 62.0 | **91.83** | 42.16 | 6.31 | 99.5 | 44.87 | 42.45 |
| | SnapKV | 28.39 | 42.19 | 53.74 | 55.42 | 43.93 | 30.44 | **29.86** | 24.51 | 23.86 | **70.5** | 91.74 | 42.46 | 6.31 | 99.5 | 52.22 | 48.64 |
| | + OBCACHE-VALUE | 29.19 | 41.86 | 53.95 | **55.43** | 44.54 | 29.94 | 29.76 | **24.77** | 23.64 | 69.5 | 91.73 | 43.18 | 6.31 | 99.5 | 52.06 | 49.04 |
| | + OBCACHE-KEY | **29.53** | **42.71** | **54.81** | 55.24 | **44.65** | 30.13 | 29.85 | 24.37 | **24.11** | 69.5 | **91.91** | 42.9 | 6.31 | 99.5 | 52.14 | 48.63 |
| | + OBCACHE-JOINT | 28.78 | 40.9 | 54.08 | 55.25 | 44.34 | **30.51** | 29.79 | 24.33 | 23.87 | 69.5 | 91.56 | **43.2** | 6.56 | 99.5 | **52.55** | **49.19** |

## D.3    LONGBENCH FULL TABLES

Due to space constraints, we present the full table results of LongBench in Table 4 and Table 5.

## E    IMPLEMENTATION OF OBCACHE

We provide a code implementation of OBCACHE in pseudo PyTorch style, as illustrated in Algorithm 1 and Algorithm 2. These two algorithms demonstrate the computation of OBCACHE saliency scores and the cache eviction operation in the prefill phase.

Table 5: LongBench results for Qwen-2.5-7B-Instruct.

| Methods | Single-Document QA | | | Multi-Document QA | | | Summarization | | | Few-shot Learning | | | Synthetic | | Code | |
|---|---|---|---|---|---|---|---|---|---|---|---|---|---|---|---|---|
| | NrtvQA | Qasper | MF-en | HotpotQA | 2WikiMQA | Musique | GovReport | QMSum | MultiNews | TREC | TriviaQA | SAMSum | PCount | PRe | Lcc | RB-P |
| ALL KV | 29.16 | 43.24 | 52.94 | 58.87 | 48.15 | 31.06 | 32.65 | 23.65 | 24.22 | 72.5 | 89.28 | 45.51 | 8.5 | 100.0 | 58.23 | 65.28 |
| **5% KV** | | | | | | | | | | | | | | | | |
| H2O | 27.99 | 32.24 | 42.15 | 52.91 | 43.3 | 27.25 | 24.61 | 22.59 | 17.84 | 36.0 | 87.06 | 44.32 | 9.0 | 100.0 | 45.61 | 56.48 |
| + OBCACHE-VALUE | **28.32** | **33.12** | 43.53 | 52.7 | 42.62 | 25.68 | 24.98 | 22.82 | 18.33 | 36.0 | 86.9 | 44.0 | 9.0 | 100.0 | **45.63** | 57.01 |
| + OBCACHE-KEY | 28.2 | 32.49 | 42.91 | **53.82** | 42.62 | 26.88 | **25.27** | 22.76 | **18.38** | 36.5 | 87.38 | 43.78 | 9.0 | 100.0 | 44.57 | **57.06** |
| + OBCACHE-JOINT | 28.15 | 32.53 | **45.02** | 53.52 | **43.94** | **28.2** | 25.1 | **23.01** | 18.28 | 36.0 | **87.74** | **44.4** | 9.0 | 100.0 | 45.59 | 56.95 |
| TOVA | 21.07 | 26.32 | **36.87** | 51.0 | 38.78 | 29.6 | 23.7 | 21.14 | 17.09 | 36.0 | 87.0 | 44.8 | 9.0 | 100.0 | 41.09 | 45.24 |
| + OBCACHE-VALUE | 23.0 | 27.32 | 35.91 | 51.66 | 40.14 | 29.01 | 23.88 | **21.43** | 17.13 | 39.5 | 87.07 | 44.58 | 9.0 | 100.0 | 44.03 | 46.58 |
| + OBCACHE-KEY | 22.93 | **27.52** | 35.06 | **51.67** | **40.27** | 29.52 | 23.91 | 21.27 | **17.26** | 41.5 | 87.05 | 44.76 | 9.0 | 100.0 | 43.06 | **46.88** |
| + OBCACHE-JOINT | **23.24** | 27.09 | 36.47 | 51.18 | 39.53 | **29.7** | **24.07** | 21.19 | 17.22 | 39.0 | **87.49** | **44.89** | 9.0 | 100.0 | **44.1** | 46.39 |
| SnapKV | 29.19 | 36.57 | **51.76** | 55.6 | **46.56** | 30.23 | 25.09 | **22.95** | 17.23 | 68.0 | 87.37 | 43.69 | 9.0 | 100.0 | 45.98 | 58.74 |
| + OBCACHE-VALUE | 28.96 | **37.73** | 51.36 | 55.55 | 45.94 | **30.98** | 25.14 | 22.75 | 17.49 | 68.5 | **87.6** | 43.35 | 9.0 | 100.0 | 46.8 | 59.42 |
| + OBCACHE-KEY | **29.9** | 36.88 | 51.05 | 55.08 | 45.36 | 30.47 | **25.37** | 22.42 | 17.4 | 65.0 | 87.41 | 43.43 | 9.0 | 100.0 | **47.88** | **59.72** |
| + OBCACHE-JOINT | 28.82 | 36.84 | 49.92 | **55.65** | 45.5 | 30.59 | 25.29 | 22.2 | **17.71** | 66.5 | 86.23 | **44.32** | 9.0 | 100.0 | 47.62 | 59.63 |
| **10% KV** | | | | | | | | | | | | | | | | |
| H2O | 28.48 | **36.71** | 45.76 | 52.66 | 43.73 | 28.8 | 26.54 | 22.84 | 19.2 | 39.5 | 88.34 | 45.09 | 9.0 | 100.0 | 53.24 | **61.41** |
| + OBCACHE-VALUE | 28.99 | 36.28 | **46.95** | 53.93 | 44.89 | 29.2 | 27.27 | 22.75 | 19.65 | 39.5 | 88.5 | 45.34 | 9.0 | 100.0 | 53.81 | 61.05 |
| + OBCACHE-KEY | 29.13 | 35.37 | 46.24 | **55.02** | **46.21** | 29.4 | 27.49 | **22.97** | 19.62 | 39.0 | 88.86 | 44.5 | 9.0 | 100.0 | **54.17** | 61.23 |
| + OBCACHE-JOINT | **29.23** | 36.46 | 46.43 | 55.01 | 44.16 | 29.06 | 27.04 | 22.74 | **19.74** | 40.0 | **89.08** | **45.43** | 9.0 | 100.0 | 53.72 | 61.1 |
| TOVA | 25.96 | 31.33 | 41.37 | 52.9 | 42.87 | **30.31** | 26.15 | 21.84 | 18.34 | 54.0 | 87.13 | 45.24 | 9.0 | 100.0 | 45.49 | 48.63 |
| + OBCACHE-VALUE | 27.28 | **32.21** | 41.26 | 53.36 | **44.01** | 29.48 | 26.21 | **21.93** | 18.4 | **57.0** | 87.03 | **45.71** | 9.0 | 100.0 | 45.35 | 48.77 |
| + OBCACHE-KEY | 26.2 | 30.62 | **43.15** | **53.87** | 41.72 | 29.39 | **26.34** | 21.82 | **18.76** | 56.0 | 87.13 | 45.46 | 9.0 | 100.0 | **45.99** | **51.34** |
| + OBCACHE-JOINT | 27.07 | 31.65 | 41.02 | 53.54 | 42.75 | 29.26 | **26.34** | **21.93** | 18.42 | 55.0 | **87.16** | 45.66 | 9.0 | 100.0 | 45.75 | 49.94 |
| SnapKV | 28.43 | 38.88 | 50.3 | **57.49** | 46.91 | 31.17 | 26.91 | 23.02 | 18.99 | 68.5 | 88.72 | 44.93 | 9.0 | 100.0 | 54.55 | 62.2 |
| + OBCACHE-VALUE | 28.16 | 38.38 | 50.96 | 56.47 | **47.12** | **31.96** | 27.25 | **23.11** | **19.36** | 69.0 | 88.28 | 44.86 | 9.0 | 100.0 | 54.74 | 61.79 |
| + OBCACHE-KEY | **29.4** | 38.92 | 51.09 | 57.06 | 45.69 | 30.76 | 27.22 | 22.87 | 19.31 | 68.5 | 88.92 | 44.51 | 9.0 | 100.0 | **54.94** | 62.28 |
| + OBCACHE-JOINT | 29.21 | **39.87** | **51.37** | 57.21 | 45.83 | 31.4 | 27.15 | 22.96 | 19.06 | 68.0 | **89.24** | **45.0** | 9.0 | 100.0 | 54.82 | **62.31** |
| **20% KV** | | | | | | | | | | | | | | | | |
| H2O | **30.08** | 37.93 | 49.47 | 55.56 | 46.0 | 29.78 | 28.83 | 23.29 | 20.73 | 45.0 | 89.35 | 45.16 | 9.0 | 100.0 | 56.4 | 63.34 |
| + OBCACHE-VALUE | 29.41 | 39.05 | 49.72 | 55.99 | **47.33** | 30.15 | **29.42** | 22.93 | 20.88 | 46.0 | 88.93 | 45.69 | 9.0 | 100.0 | 56.81 | 63.39 |
| + OBCACHE-KEY | 29.06 | 38.79 | 48.61 | 55.62 | 46.1 | **30.37** | 29.06 | **23.31** | **21.34** | 48.0 | 89.48 | 45.71 | 9.0 | 100.0 | 56.98 | **63.6** |
| + OBCACHE-JOINT | 29.46 | **39.2** | **50.15** | 56.45 | 46.67 | 30.02 | 29.29 | 22.99 | 20.93 | 47.5 | 89.32 | 45.61 | 9.0 | 100.0 | **57.48** | 63.5 |
| TOVA | 28.67 | 35.62 | 47.05 | 55.38 | 42.75 | 29.24 | 29.04 | 22.56 | 20.39 | 66.0 | 88.7 | 45.82 | 9.0 | 100.0 | 49.63 | 53.97 |
| + OBCACHE-VALUE | **28.75** | 35.94 | 47.48 | 56.03 | 44.35 | 29.57 | **29.31** | 22.74 | 20.21 | 67.0 | **89.51** | 45.69 | 9.0 | 100.0 | 49.79 | 55.29 |
| + OBCACHE-KEY | 28.34 | 36.14 | 47.46 | **56.29** | **44.8** | **30.06** | 28.97 | 22.32 | **20.62** | 67.0 | 89.15 | 45.23 | 9.0 | 100.0 | 50.23 | **55.5** |
| + OBCACHE-JOINT | 28.5 | **36.86** | **47.81** | 56.18 | 44.66 | 29.6 | 29.0 | **22.85** | 20.29 | 66.0 | 89.01 | **45.84** | 9.0 | 100.0 | **50.51** | 55.33 |
| SnapKV | **29.67** | 41.08 | 50.98 | **57.92** | **46.97** | 31.02 | 29.07 | 23.04 | 20.53 | **71.0** | 89.09 | 45.25 | 9.0 | 100.0 | 57.18 | 63.32 |
| + OBCACHE-VALUE | 28.59 | 40.77 | 51.46 | 57.64 | 46.7 | 31.13 | 29.22 | 23.22 | 20.64 | 70.0 | 89.09 | 44.84 | 9.0 | 100.0 | 57.13 | 63.54 |
| + OBCACHE-KEY | 29.26 | 40.19 | **51.72** | 57.34 | 46.03 | **31.37** | 29.32 | 23.33 | **21.02** | 70.0 | 88.99 | 44.86 | 9.0 | 100.0 | 57.53 | 63.7 |
| + OBCACHE-JOINT | 29.38 | **41.14** | 51.17 | 57.39 | 45.82 | **31.37** | **29.46** | **23.36** | 20.61 | 70.5 | **89.15** | 45.11 | 9.0 | 100.0 | **57.55** | **63.78** |

**Algorithm 1** Implementation of OBCACHE score update in pseudo PyTorch style.

```python
# key_states/value_states: cache matrix (bsz, num_heads, kv_len, head_dim);
# A/Z: attention weight/logit matrix (bsz, num_heads, q_len, kv_len);
# O: attention output matrix (bsz, num_heads, q_len, head_dim);
# w: perturbation window start index;

def obcache_score(key_states, value_states, A, Z, O, w):
    # Target only the perturbation window
    A = A[..., -w:, :]
    Z = Z[..., -w:, :]
    O = O[..., -w:, :]

    # Compute accumulated attention-based score
    S_attn_based = A.pow(2).sum(-2)
    ### Existing cache eviction scores (e.g., H2O and TOVA) end here ###

    # Compute value-pruning score
    V_2norm = value_states.pow(2).sum(dim=-1)
    S_value = S_attn * V_2norm

    # Compute key-pruning score
    O_2norm = O.pow(2).sum(dim=-1)
    VO = torch.einsum('bhqd,bhpd->bhqp', O, V)
    VmO_2norm = O_2norm.unsqueeze(-1) + V_2norm.unsqueeze(-2) - 2 * VO
    S_key = ((A * Z).pow(2) * VmO_2norm).sum(dim=-2)

    # Compute joint-pruning score
    VVmO = V_2norm.unsqueeze(-2) - VO
    S_joint = (2 * A.pow(2) * Z * VVmO).sum(dim=-2) + S_value + S_key

    return S_attn_based, S_value, S_key, S_joint
```

**Algorithm 2** Implementation of OBCACHE cache eviction in pseudo PyTorch style.

```python
# S: cache eviction saliency score matrix (bsz, num_heads, kv_len);
# num_hh / num_recent: cache budget allocated for recent and heavy-hitter tokens;
# num_coming: number of maximum tokens to come in next step;

def evict_kv(S, key_states, value_states, num_hh, num_recent, num_coming):
    bsz, num_heads, kv_len, head_dim = value_states.shape
    cutoff = kv_len - num_recent + num_coming

    # Select most salient token positions
    keep_topk_idx = torch.topk(S[..., :cutoff], num_hh, dim=-1)[1].sort().values
    keep_topk_idx = keep_topk_idx.unsqueeze(-1).expand(-1, -1, -1, head_dim)

    # Evict and keep recent
    k_hh = torch.gather(key_states[..., :cutoff, :], dim=-2, index=keep_topk_idx)
    k_compress = torch.cat([k_hh, key_states[..., cutoff:, :]], dim=-2)

    v_hh = torch.gather(value_states[..., :cutoff, :], dim=-2, index=keep_topk_idx)
    v_compress = torch.cat([v_hh, value_states[..., cutoff:, :]], dim=-2)

    return k_compress, v_compress
```

