# OpenReview forum: "OBCache: Optimal Brain KV Cache Pruning for Efficient Long-Context LLM Inference"
_ICLR.cc/2026/Conference — Submitted to ICLR 2026_

### Official Review · Reviewer_P3QT · 2025-10-16

**Soundness:** 2
**Presentation:** 3
**Contribution:** 2
**Rating:** 2
**Confidence:** 5

**Summary:**

This paper introduces OBCache, which formulates the KV cache eviction process as a pruning problem. Through an analysis based on output perturbation, they enhance existing scoring metrics that rely on attention weights. By integrating OBCache, current methods can achieve higher accuracy by better determining which tokens to discard.

**Strengths:**

1. The paper is well-written and easy to follow.

**Weaknesses:**

1. The analysis based on output perturbation has been explored in prior work, both in KV cache eviction  [1] and merge [2]. Notably, [1] has already pointed out that attention weights alone are not a sufficient scoring metric in kv cache eviction. Furthermore, the empirical finding that the value norm serves as a good indicator has also been proposed [3].  A more detailed discussion is needed to differentiate the theoretical analysis and empirical results from these existing studies.

[1] Feng, Yuan, et al. "Identify critical kv cache in llm inference from an output perturbation perspective." arXiv preprint arXiv:2502.03805 (2025).

[2] Tian, Yuxuan, et al. "KeepKV: Eliminating Output Perturbation in KV Cache Compression for Efficient LLMs Inference." arXiv preprint arXiv:2504.09936 (2025).

[3] Guo, Zhiyu, et ak. "Attention score is not all you need for token importance indicator in kv cache reduction: Value also matters." arXiv preprint arXiv:2406.12335 (2024).

2. The improvement brought by OBCache appears to be limited, especially on the SnapKV baseline. Given that SnapKV is the SOTA method compared to H2O and TOVA, it is unclear whether the gains offered are significant enough to advance the state of the art in this field.

3. The computational overhead introduced by the improved scoring metric itself is not evaluated. I suggest that the authors provide a detailed analysis of this overhead.

4. The paper's evaluation focuses heavily on high compression ratios where performance degradation is substantial. For practical applications, it would be more instructive to focus the evaluation on a realistic range of performance loss (e.g., less than 5%) to better guide real-world implementation.

5. The experimental setup described in the appendix contains an inconsistency in the perturbation window size. A fixed size of 16 is used for SnapKV and H2O on the NIAH tasks, whereas a dynamic size of 5% of the prompt length is used for LongBench. This raises questions about the sensitivity to this hyperparameter. It is important to clarify how this window size should be set in practical scenarios and justify the chosen settings.

**Questions:**

See weaknesses.

---

> ### Author Response · Authors · 2025-11-25
> **Response (1/3)**
>
> Thank you for reviewing our paper and providing constructive feedbacks. In response to your comments, we address your concerns as outlined below:
>
> > W1: Discussion and comparison with relevant work.
>
> Thank you for pointing out the relevance of [1–3]. We apologize for not including them in our original submission. We were not aware of these works when conducting our study because they are very recent. We have now carefully reviewed them and appreciate the opportunity to clarify the differences.
>
> While both [1, 2] and our work use output perturbation as a measure of token saliency, the core theoretical foundation and resulting methodology differ substantially. We want to clarify that output perturbation is a standard objective in layer-wise pruning, and within KV cache it has also been explored since CaM [4]. Our novelty is not in adopting this objective alone, but in formulating KV cache eviction as a structured pruning problem based on Optimal Brain Damage (OBD), which to the best of our knowledge has not been done previously. Furthermore, our theoretical analysis based on OBD therefore differs fundamentally from [1].
>
> Specifically, our analysis closely follows the OBD framework widely used in model pruning, which requires a Taylor expansion of the output perturbation with respect to the pruning units. Because the pruning units in KV cache are dynamic key and value matrices rather than static model parameters, deriving both first- and second-order information requires new theoretical effort, ultimately yielding three saliency scores (for key-, value-, and joint-pruning).
>
> In contrast, [1] derives an upper bound on output perturbation, leading to a closed-form expression depending only on $\mathbf{A}\mathbf{V}$. Consequently, [1] obtains a score equivalent to only one component of our results (the value-pruning score). Its method also requires a two-stage algorithm whose first step still selects tokens using attention weights, whereas our scores directly replace attention-weight heuristics under the structured pruning formulation. Moreover, [1] defines perturbation only on the current-step attention output, which does not extend to decoding-stage eviction, and does not consider the role of key matrices, which we show to be crucial. Similarly, [3] empirically identifies a score equivalent to our value-pruning score, but lacks a theoretical formulation. [2] analyzes KV cache merging based on current-step attention output, but does not use any OBD-based structured pruning analysis, and therefore differs in both scope and methodology.
>
> A key strength of our pruning-based formulation is its flexibility in choosing both the objective function and pruning variables. For example, our analysis does not assume a fixed sequence dimension for $\mathbf{O}$: it may represent a partial output (i.e., $\mathbf{O}_{w:s}$) or expand as decoding progresses. This enables both prefill- and decoding-stage pruning, naturally yielding the perturbation window that unifies differences among H2O, TOVA, and SnapKV (Section 3.4). Furthermore, while not explored in this work, our framework can be extended to unstructured or channel-wise pruning by redefining the pruning variables, or to cache merging by relaxing the diagonal Hessian assumption. Both are standard directions in model pruning literature. Therefore, our theoretical framework directly connects KV cache compression to model pruning and provides a principled foundation for future extensions.
>
> We hope this discussion clearly differentiates our work from [1-3] and will clarify these distinctions in the revised manuscript.
>
> [1] Identify Critical KV Cache in LLM Inference from an Output Perturbation Perspective. Feng et al., 2025.
>
> [2] KeepKV: Eliminating Output Perturbation in KV Cache Compression for Efficient LLMs Inference. Tian et al., 2025
>
> [3] Attention score is not all you need for token importance indicator in kv cache reduction: Value also matters. Guo et al., 2024.
>
> [4] CaM: Cache Merging for Memory-efficient LLMs Inference. Zhang et al., 2024.

---

> > ### Author Response · Authors · 2025-11-25
> > **Response (2/3)**
> >
> > > W2: Improvement brought by OBCache appears to be limited, especially on the SnapKV baseline.
> >
> > Thank you for the observation. We would first like to clarify that the primary contribution of this work is the structured-pruning framework that formulates KV cache eviction from a theoretically grounded perspective, generalizing prior attention-weight scoring methods. This framework directly connects dynamic KV cache compression to model pruning and enables promising extensions.
> >
> > The empirical evaluation is intended to validate that OBCache scores (derived from an attention-output-level objective) are more effective than purely attention-weight scores (derived from a reduced attention-matrix-level objective). This is already supported by the consistent improvements observed on H2O and TOVA (e.g., over 10% accuracy gain on NIAH and ~0.5 perplexity reduction on PG19), both of which rely solely on attention weights for eviction.
> >
> > The smaller improvement observed on SnapKV is expected. SnapKV applies an additional pooling layer to smooth token scores, which is an auxiliary heuristic not captured by our framework. This step may interfere with the benefits of more accurate saliency estimates from OBCache, limiting performance gains when simply replacing the scoring component. Our current implementation does not modify or optimize this pooling mechanism, and exploring compatibility between OBCache and such heuristic components remains an open direction for future work.
> >
> > In summary, the improvements on H2O and TOVA already provide strong evidence that OBCache scores outperform purely attention-weight scoring. Exploring better integration with state-of-the-art methods is not the focus of this work but represents a promising direction for further extension.
> >
> > > W3: Evaluation of OBCache's computation overhead.
> >
> > We have added a detailed discussion in Appendix D.1 of the revised manuscript. We first provide a theoretical comparison of the runtime complexity of OBCache scores versus attention-based baselines:
> >
> > [W3-1]
> > | Method         | Complexity |
> > |--------------- |------------|
> > | Attn. Based    | $O(W)$|
> > | OBCache-Value  | $O(W + d_{\mathrm{hidden}})$|
> > | OBCache-Key    | $O(W d_{\mathrm{hidden}})$|
> > | OBCache-Joint  | $O(W d_{\mathrm{hidden}})$|
> >
> > Here, $W$ denotes the perturbation window size and $d_{\mathrm{hidden}}$ represents the per-head hidden dimension. Computing the value-pruning score only adds a small linear term in $d_{\mathrm{hidden}}$, whereas key- and joint-pruning require computing the norm of difference between each value vector and its corresponding attention output within the window, resulting in multiplicative complexity. However, in practical settings, $W$ is very small (e.g., $W=16$ in prefill and $W=1$ in decoding), so the additional cost remains minor. Additionally, the computations within the window are independent and inherently parallelizable, enabling efficient batched GPU execution that further mitigates the multiplicative factor in practice.
> >
> > We further benchmark the actual runtime of OBCache during both static and dynamic cache eviction, comparing prefill and decoding latency to attention-based baselines across different context lengths on a single A100-80GB GPU with batch size 1 (batch size 4 is reported in the revised manscript).
> >
> > [W3-2] Decoding (ms/token)
> > | Method         | 32k | 64k | 96k | 128k |
> > |--------------- |-----|-----|-----|------|
> > | Full KV        |68.66|133.2|192.5|OOM  |
> > | Attn. Based    |55.39|63.04|62.05|62.79|
> > | OBCache-Value  |56.45|62.93|63.35|64.82|
> > | OBCache-Key    |65.25|71.68|72.67|72.28|
> > | OBCache-Joint  |70.53|76.77|77.19|76.43|
> >
> > In decoding, we let the model generate 512 steps with varying context sizes, dynamically updating scores and evicting cache tokens at every step. We find that value-pruning score introduces negligible overhead (< 2 ms/token) relative to attention-based methods, and key- and joint-pruning scores introduce < 15 ms/token across all context lengths. Although the latter two are more expensive, they remain minor compared to the full-KV baseline as the context increases.
> >
> > [W3-3] Prefill (s)
> > | Method         | 32k | 64k | 96k | 128k |
> > |--------------- |-----|-----|-----|-----|
> > | Full KV        |3.556|9.634|18.44|30.02|
> > | Attn. Based    |3.654|9.903|19.03|30.72|
> > | OBCache-Value  |3.670|9.967|19.04|30.83|
> > | OBCache-Key    |3.698|10.02|19.10|30.90|
> > | OBCache-Joint  |3.732|10.04|19.15|30.95|
> >
> > In prefill, we measure the average seconds required from the start of prefill until the first token is generated, including the cache eviction operation. In this case, all eviction methods show negligible additional latency compared to the full-KV baseline. These empirical results align with the complexity analysis and demonstrate that OBCache scores can be efficiently computed.

---

> > > ### Author Response · Authors · 2025-11-25
> > > **Response (3/3)**
> > >
> > > > W4: Evaluation focuses heavily on high compression ratios where performance degradation is substantial. It would be more instructive to focus the evaluation on a realistic range of performance loss (e.g., less than 5%).
> > >
> > > We would like to clarify that many of the SnapKV-relevant settings in our evaluation, such as 320 and 400 KV budgets on passkey retrieval and 5× and 10× compression rates on LongBench, already fall within a <5% performance loss range relative to the full-KV baseline. To maintain a fair comparison at the same compression rates, we also evaluate H2O and TOVA in these regimes, where they exhibit larger performance degradation
> > >
> > > To further address your concern, we conduct additional experiments for H2O and TOVA at lower compression rates (1600 KV budget, corresponds to 20% KV budget) on the passkey retrieval task (8K).
> > >
> > > | Method         | NIAH (8K) w/ 20% KV|
> > > |--------------- |---------|
> > > | H2O              | 95.2 |
> > > | + OBCache-Value  | 95.6 |
> > > | + OBCache-Key    | 96.0 |
> > > | + OBCache-Joint  | 96.4 |
> > > | TOVA             | 96.8 |
> > > | + OBCache-Value  | 97.2 |
> > > | + OBCache-Key    | 97.2 |
> > > | + OBCache-Joint  | 98.0 |
> > >
> > > These results, together with those in the main paper, demonstrate that OBCache scores surpass attention weights across both low-compression and high-compression regimes, and the improvements become more evident under challenging high-compression settings.
> > >
> > >
> > > > W5: Inconsistent experimental setup in the perturbation window size and clarification on how the window size should be set in practical scenarios.
> > >
> > > The different perturbation window setups arise from the characteristics of the two benchmark types. NIAH is a synthetic dataset in which all samples share the same prompt length, and the information required to answer the question (the passkey) has a fixed length. Therefore, a constant cache budget and a fixed perturbation window are appropriate. In contrast, LongBench contains tasks with variable prompt lengths and varying amount of task-relevant information, so we adopt a ratio-based window size, which scales naturally across instances.
> > >
> > > As discussed in Section 3.4, prior attention-based cache eviction methods accumulate attention scores across different query positions. Under our structured pruning framework, this is equivalent to accumulating output perturbation over different query positions, which naturally leads to the concept of a perturbation window. OBCache provides plug-and-play scoring modules that directly replace attention-weight scores while preserving each prior method's original accumulation strategy. Therefore, the perturbation window is not a newly introduced hyperparameter, but **rather a unifying notion** that connects and formalizes the accumulation strategies of prior methods.
> > >
> > > We also provided empirical insights on the window size in Section 3.5. When the perturbation window is small, selected tokens align closely with the oracle ground truth. As the window grows, early tokens accumulate disproportionately high scores, thereby reducing selection performance. In such cases, enforcing a recent-retention window, as in H2O, can improve performance. When the window is in a small range, performance differences are minor. For example, SnapKV conducts an ablation study on what they call the "observation window" (which corresponds to the perturbation window under our framework) and reports similar LongBench performance when this window is set to 16, 32, or 64.

---

### Official Review · Reviewer_AP9w · 2025-10-17

**Soundness:** 3
**Presentation:** 3
**Contribution:** 3
**Rating:** 6
**Confidence:** 3

**Summary:**

This paper proposes **OBCache**, a principled framework that formulates the KV-cache eviction task as a structured pruning problem inspired by the Optimal Brain Damage/Surgeon paradigm. Instead of relying on heuristic accumulation of attention weights, the method estimates each token’s true contribution by approximating its second-order effect on the loss function. Experiments across multiple LLMs and long-context benchmarks demonstrate strong empirical gains, with OBCache preserving model fidelity under tight memory budgets. The paper is clearly written, well-motivated, and presents a meaningful advance in cache management for long-context inference.

**Strengths:**

The key strength lies in the elegant adaptation of classic pruning theory to dynamic KV management. The proposed importance metric provides a theoretically grounded and interpretable measure of token salience, effectively bridging a gap between structured pruning and attention-based heuristics. Implementation is straightforward and modular, and the method delivers consistent improvements in perplexity and zero-shot accuracy across different architectures. The ablations and visualizations are thorough and clearly communicate the method’s behavior.

**Weaknesses:**

1. While the theoretical formulation is sound, the **practical efficiency** of OBCache remains unclear. The computation of second-order–inspired importance scores is nontrivial, yet the paper does not provide a quantitative breakdown of end-to-end inference latency or throughput compared to lighter heuristics such as H2O or StreamingLLM. A latency–quality trade-off analysis would strengthen the empirical credibility of the work.

2. Since the importance score relies on a local second-order approximation, its **long-term stability** over very long contexts (e.g., 100k+ tokens) is not well discussed. It would be valuable to analyze whether cumulative approximation error leads to score drift or suboptimal pruning in extended sequences.

3. The **layer-wise pruning strategy** raises a practical question: is there a global control mechanism (e.g., a unified sparsity target) that governs cache size across all layers, or must each layer’s threshold be tuned manually? Clarifying this would improve the method’s usability for large-scale deployment.

**Questions:**

1. Could the authors quantify the per-token latency or throughput overhead introduced by OBCache on standard hardware (e.g., A100/H100) relative to attention-based heuristics?

2. How stable are the OBCache importance scores over extremely long contexts? Have the authors observed cumulative error or bias toward earlier tokens?

3. Is there a simple way to enforce a global cache-budget constraint across layers, rather than tuning each layer’s threshold individually?

---

> ### Author Response · Authors · 2025-11-25
> **Response (1/2)**
>
> Thank you for reviewing our paper and providing constructive feedbacks. In response to your comments, we address your concerns as outlined below:
>
> > W1, Q1: Overhead introduced by computing OBCache scores compared to purely attention-based methods.
>
> We have added a detailed discussion in Appendix D.1 of the revised manuscript. We first provide a theoretical comparison of the runtime complexity of OBCache scores versus attention-based baselines:
>
> [W1-1]
> | Method         | Complexity |
> |--------------- |------------|
> | Attn. Based    | $O(W)$|
> | OBCache-Value  | $O(W + d_{\mathrm{hidden}})$|
> | OBCache-Key    | $O(W d_{\mathrm{hidden}})$|
> | OBCache-Joint  | $O(W d_{\mathrm{hidden}})$|
>
> Here, $W$ denotes the perturbation window size and $d_{\mathrm{hidden}}$ represents the per-head hidden dimension. Computing the value-pruning score only adds a small linear term in $d_{\mathrm{hidden}}$, whereas key- and joint-pruning require computing the norm of difference between each value vector and its corresponding attention output within the window, resulting in multiplicative complexity. However, in practical settings, $W$ is very small (e.g., $W=16$ in prefill and $W=1$ in decoding), so the additional cost remains minor. Additionally, the computations within the window are independent and inherently parallelizable, enabling efficient batched GPU execution that further mitigates the multiplicative factor in practice.
>
> We further benchmark the actual runtime of OBCache during both static and dynamic cache eviction, comparing prefill and decoding latency to attention-based baselines across different context lengths on a single A100-80GB GPU with batch size 1 (batch size 4 is reported in the revised manscript).
>
> [W1-2] Decoding (ms/token)
> | Method         | 32k | 64k | 96k | 128k |
> |--------------- |-----|-----|-----|------|
> | Full KV        |68.66|133.2|192.5|OOM  |
> | Attn. Based    |55.39|63.04|62.05|62.79|
> | OBCache-Value  |56.45|62.93|63.35|64.82|
> | OBCache-Key    |65.25|71.68|72.67|72.28|
> | OBCache-Joint  |70.53|76.77|77.19|76.43|
>
> In decoding, we let the model generate 512 steps with varying context sizes, dynamically updating scores and evicting cache tokens at every step. We find that value-pruning score introduces negligible overhead (< 2 ms/token) relative to attention-based methods, and key- and joint-pruning scores introduce < 15 ms/token across all context lengths. Although the latter two are more expensive, they remain minor compared to the full-KV baseline as the context increases.
>
> [W1-3] Prefill (s)
> | Method         | 32k | 64k | 96k | 128k |
> |--------------- |-----|-----|-----|-----|
> | Full KV        |3.556|9.634|18.44|30.02|
> | Attn. Based    |3.654|9.903|19.03|30.72|
> | OBCache-Value  |3.670|9.967|19.04|30.83|
> | OBCache-Key    |3.698|10.02|19.10|30.90|
> | OBCache-Joint  |3.732|10.04|19.15|30.95|
>
> In prefill, we measure the average seconds required from the start of prefill until the first token is generated, including the cache eviction operation. In this case, all eviction methods show negligible additional latency compared to the full-KV baseline. These empirical results align with the complexity analysis and demonstrate that OBCache scores can be efficiently computed.

---

> > ### Author Response · Authors · 2025-11-25
> > **Response (2/2)**
> >
> > > W2, Q2: Evaluation at ultra-long contexts and accumulative errors.
> >
> > Thank you for pointing this out. We provide additional results on the passkey retrieval task with context lengths scaled up to 128k using 400 KV budget (with 80 synthetic samples), which is shown in the table below. The result shows that OBCache scores consistently outperform their purely attention-based counterparts (H2O and TOVA) in extremely long contexts.
> >
> > | Method         | 128K Context |
> > |--------------- |---------|
> > | H2O              | 23.2 |
> > | + OBCache-Value  | 31.6 |
> > | + OBCache-Key    | 31.2 |
> > | + OBCache-Joint  | 33.6 |
> > | TOVA             | 31.2 |
> > | + OBCache-Value  | 38.0 |
> > | + OBCache-Key    | 42.8 |
> > | + OBCache-Joint  | 40.4 |
> >
> > This suggests that OBCache does not accumulate errors over time and remains robust as context length grows. Instead, a common challenge in layer-wise pruning frameworks is the gradual accumulation of approximation errors across layers, which may eventually degrade performance. We see this as a future work, where OBCache could be further enhanced to improve performance.
> >
> > > W3, Q3: Is there a global control mechanism that governs cache size across all layers, rather than tuning each layer’s threshold individually?
> >
> > Thank you for the insightful question. Although we formulate cache eviction as a layer-wise pruning problem, in our experiments we adopt a uniform cache budget across layers, which enforces an $L_0$-style sparsity constraint without tuning per-layer thresholds individually.
> >
> > To support non-uniform allocation, one could compute the empirical CDF of saliency scores across all layers and apply a global quantile threshold, which would automatically induce non-uniform per-layer budgets based on their score distributions.
> >
> > Additionally, several recent works have studied non-uniform KV allocation across layers [1] or across attention heads [2]. These strategies are orthogonal to our contribution and could be combined with OBCache scores to further enhance performance.
> >
> >
> > [1] PyramidInfer: Pyramid KV cache compression for high-throughput LLM inference. Yang et al., 2024
> >
> > [2] Ada-KV: Optimizing KV Cache Eviction by Adaptive Budget Allocation for Efficient LLM Inference. Feng et al., 2025.

---

> > > ### Comment · Reviewer_AP9w · 2025-11-25
> > >
> > > The authors addressed my concerns well. The added experiments and explanations strengthen the paper. I'm maintaining my positive score.

---

### Official Review · Reviewer_UVpj · 2025-10-18

**Soundness:** 3
**Presentation:** 3
**Contribution:** 3
**Rating:** 6
**Confidence:** 3

**Summary:**

This manuscript introduces OBCache, a novel framework for optimizing inference efficiency in large language models (LLMs) with long context windows through principled key–value (KV) cache eviction. The method formulates cache eviction as a structured pruning problem grounded in Optimal Brain Damage (OBD) theory. It computes closed-form token saliency scores—value-based, key-based, and joint key–value pairs—using second-order Taylor approximations of the perturbation in attention outputs due to pruning. This framework generalizes and improves upon existing attention-weight-based heuristics (e.g., H2O, TOVA, SnapKV). The paper provides both theoretical analysis and extensive empirical evaluation across long-context tasks (Needle-in-a-Haystack, LongBench, PG19), showing consistent improvements in performance when integrating OBCache into existing cache eviction strategies.

**Strengths:**

1. The idea to evict tokens by measuring the impact of keys, values and key-value pairs is novel. This approach better capture token importance compared to prior attention-only heuristics.
2. Experiments that cover a range of models (LLaMA-3.1, Qwen-2.5) and tasks (retrieval, QA, summarization, perplexity). Results clearly show consistent performance improvement under tight KV budgets.

**Weaknesses:**

1. The paper does not quantify the computational or memory overhead associated with computing perturbation-based saliency scores during inference.
2. The quality of the perturbation approximation is not explicitly evaluated. It would strengthen the work to analyze the accuracy of the estimation framework (e.g., how well the derived scores correlate with the true eviction-induced error). Furthermore, quantifying the upper-bound performance achievable with oracle (ground-truth) saliency would clarify the potential headroom for improvement.
3. Experiments are reported only up to 32K context length. It remains uncertain how OBCache scales to ultra-long contexts, where accumulated approximation errors may degrade performance.

**Questions:**

1. What is the runtime and memory overhead introduced by computing OBCache saliency scores compared to purely attention-based heuristics such as H2O or TOVA?
2. How accurate is the proposed perturbation-based estimation of token importance? If the true eviction impact (oracle ground truth) were used instead, what would be the maximum achievable performance? In that case, would the joint key–value pruning strategy yield the best results?
3. How does OBCache perform at very long context lengths (e.g., 64K–128K tokens)? Does the perturbation-based approximation remain stable, or does accuracy deteriorate with context depth?
4. The perturbation window is manually chosen. How do we select the window size in real-world use cases? Could the model learn to adjust this window online during inference?

---

> ### Author Response · Authors · 2025-11-25
> **Response (1/2)**
>
> Thank you for reviewing our paper and providing constructive feedbacks. In response to your comments, we address your concerns as outlined below:
>
> > W1, Q1: Overhead introduced by computing OBCache scores compared to purely attention-based methods.
>
> We have added a detailed discussion in Appendix D.1 of the revised manuscript. We first provide a theoretical comparison of the runtime complexity of OBCache scores versus attention-based baselines:
>
> [W1-1]
> | Method         | Complexity |
> |--------------- |------------|
> | Attn. Based    | $O(W)$|
> | OBCache-Value  | $O(W + d_{\mathrm{hidden}})$|
> | OBCache-Key    | $O(W d_{\mathrm{hidden}})$|
> | OBCache-Joint  | $O(W d_{\mathrm{hidden}})$|
>
> Here, $W$ denotes the perturbation window size and $d_{\mathrm{hidden}}$ represents the per-head hidden dimension. Computing the value-pruning score only adds a small linear term in $d_{\mathrm{hidden}}$, whereas key- and joint-pruning require computing the norm of difference between each value vector and its corresponding attention output within the window, resulting in multiplicative complexity. However, in practical settings, $W$ is very small (e.g., $W=16$ in prefill and $W=1$ in decoding), so the additional cost remains minor. Additionally, the computations within the window are independent and inherently parallelizable, enabling efficient batched GPU execution that further mitigates the multiplicative factor in practice.
>
> We further benchmark the actual runtime of OBCache during both static and dynamic cache eviction, comparing prefill and decoding latency to attention-based baselines across different context lengths on a single A100-80GB GPU with batch size 1 (batch size 4 is reported in the revised manscript).
>
> [W1-2] Decoding (ms/token)
> | Method         | 32k | 64k | 96k | 128k |
> |--------------- |-----|-----|-----|------|
> | Full KV        |68.66|133.2|192.5|OOM  |
> | Attn. Based    |55.39|63.04|62.05|62.79|
> | OBCache-Value  |56.45|62.93|63.35|64.82|
> | OBCache-Key    |65.25|71.68|72.67|72.28|
> | OBCache-Joint  |70.53|76.77|77.19|76.43|
>
> In decoding, we let the model generate 512 steps with varying context sizes, dynamically updating scores and evicting cache tokens at every step. We find that value-pruning score introduces negligible overhead (< 2 ms/token) relative to attention-based methods, and key- and joint-pruning scores introduce < 15 ms/token across all context lengths. Although the latter two are more expensive, they remain minor compared to the full-KV baseline as the context increases.
>
> [W1-3] Prefill (s)
> | Method         | 32k | 64k | 96k | 128k |
> |--------------- |-----|-----|-----|-----|
> | Full KV        |3.556|9.634|18.44|30.02|
> | Attn. Based    |3.654|9.903|19.03|30.72|
> | OBCache-Value  |3.670|9.967|19.04|30.83|
> | OBCache-Key    |3.698|10.02|19.10|30.90|
> | OBCache-Joint  |3.732|10.04|19.15|30.95|
>
> In prefill, we measure the average seconds required from the start of prefill until the first token is generated, including the cache eviction operation. In this case, all eviction methods show negligible additional latency compared to the full-KV baseline. These empirical results align with the complexity analysis and demonstrate that OBCache scores can be efficiently computed.

---

> > ### Author Response · Authors · 2025-11-25
> > **Response (2/2)**
> >
> > > W2, Q2: Maximum achievable performance by true eviction impact and quality of the perturbation approximation.
> >
> > Thank you for raising this question. As discussed in Section 3.2, the true eviction error is an unobservable quantity in practice, because the model does not have access to future query states when eviction decisions are made. Therefore, an oracle upper bound on achievable performance (i.e., selecting tokens based on the true eviction error) is not attainable in real deployment settings.
> >
> > To address this, we use the pruning-induced eviction error as a practical surrogate objective, which is measurable. We empirically demonstrate in Section 3.5 that this surrogate correlates well with the true eviction error. Specifically, we treat the top-k tokens selected according to the true eviction error as an oracle ground truth (computed exactly via brute-force forward passes evicting each token). Tokens selected using the pruning-induced eviction error (also computed exactly) achieve 85% alignment with this oracle selection, indicating strong correlation.
> >
> > Next, computing the exact pruning-induced eviction error is computationally infeasible as it requires repeated attention computation. So we derive three second-order closed-form scores that estimate it at different levels of approximation. The joint-pruning score accounts for both the individual and interactive effects of key and value pruning, and is theoretically the most accurate estimator. The value- and key-scores correspond to decomposed components that isolate the individual contributions. As shown in Figure 2, token recall using these derived scores closely matches that obtained from the exact pruning-induced eviction error, demonstrating the quality of these approximations.
> >
> > We hope this explanation clarifies the role of the true eviction error and the motivation behind our approximations.
> >
> > > W3, Q3: Evaluation at ultra-long contexts and accumulative errors.
> >
> > Thank you for pointing this out. We provide additional results on the passkey retrieval task with context lengths scaled up to 128k using 400 KV budget (with 250 synthetic samples), which is shown in the table below. The result shows that OBCache scores consistently outperform their purely attention-based counterparts (H2O and TOVA) in extremely long contexts.
> >
> > | Method         | 128K Context |
> > |--------------- |---------|
> > | H2O              | 23.2 |
> > | + OBCache-Value  | 31.6 |
> > | + OBCache-Key    | 31.2 |
> > | + OBCache-Joint  | 33.6 |
> > | TOVA             | 31.2 |
> > | + OBCache-Value  | 38.0 |
> > | + OBCache-Key    | 42.8 |
> > | + OBCache-Joint  | 40.4 |
> >
> > This suggests that OBCache does not accumulate errors over time and remains robust as context length grows. Instead, a common challenge in layer-wise pruning frameworks is the gradual accumulation of approximation errors across layers, which may eventually degrade performance. We see this as a future work, where OBCache could be further enhanced to improve performance.
> >
> > > Q4: Choice of the perturbation window and adapting it online.
> >
> > Thank you for raising the interesting question. As discussed in Section 3.4, prior attention-based cache eviction methods accumulate attention scores across different query positions. Under our structured pruning framework, this is equivalent to accumulating output perturbation over different query positions, which naturally leads to the concept of a perturbation window. OBCache provides plug-and-play scoring modules that directly replace attention-weight scores while preserving each prior method's original accumulation strategy. Therefore, the perturbation window is not a newly introduced hyperparameter, but **rather a unifying notion** that connects and formalizes the accumulation strategies of prior methods.
> >
> > We also provided empirical insights on the window size in Section 3.5. When the perturbation window is small, selected tokens align closely with the oracle ground truth. As the window grows, early tokens accumulate disproportionately high scores, thereby reducing selection performance. In such cases, enforcing a recent-retention window, as in H2O, can improve performance. When the window is in a small range, performance differences are minor. For example, SnapKV conducts an ablation study on what they call the "observation window" (which corresponds to the perturbation window under our framework) and reports similar LongBench performance when this window is set to 16, 32, or 64.
> >
> > Regarding a learnable window size, we believe this is a promising direction for future work under our framework, particularly for dynamic cache eviction. Existing dynamic eviction methods either accumulate saliency across the entire historical window (H2O) or only use the most recent perturbation (TOVA). More flexible or online-adjustable window strategies could be beneficial for streaming or multi-turn conversational settings, and we plan to explore this in future work.

---

### Official Review · Reviewer_MqZV · 2025-10-29

**Soundness:** 3
**Presentation:** 2
**Contribution:** 2
**Rating:** 4
**Confidence:** 4

**Summary:**

This paper presents Optimal Brain Cache, a theoretically grounded framework for KV cache eviction in LLMs, aiming to reduce memory and latency overheads in long-context inference. Unlike prior heuristic approaches that rely mainly on attention weights, OBCACHE formulates cache eviction as a structured pruning problem inspired by the Optimal Brain Damage theory. This formulation leads to output-aware saliency scores that integrate attention weights, value states, pre-softmax logits, and attention outputs to better reflect each token's contribution. Extensive experiments across LLaMA-3.1 and Qwen-2.5 models on long-context benchmarks, including Needle-in-a-Haystack, long-sequence perplexity, and LongBench, demonstrate that integrating OBCACHE consistently improves generation quality.

**Strengths:**

1. The paper proposes three optimization approaches for existing scoring methods based on the Optimal Brain Damage theory.
2. The method enhances the generation quality of existing cache eviction approaches in both ruler and longbench benchmark.

**Weaknesses:**

1. The theoretical presentation is poorly organized, with most details relegated to the appendix, which compromises readability. It is recommended to formalize the theoretical content using clear Theorems and Proofs to improve clarity.

2. This paper omits a closely related work [1], which similarly analyzes cache eviction from an output perturbation perspective. Notably, both papers adopt a similar objective of minimizing output perturbation and found the previous attention-weight-based methods as special cases under this formulation. A detailed discussion and comparison between the two methods would be beneficial.

3. The paper lacks an evaluation of the computational overhead introduced by the compression algorithm itself.

4. The reported improvements on real-world benchmark tasks appear marginal.

5. The paper does not provide sufficient analysis or comparison among the three variants-Value, Key, and Joint-in terms of their effectiveness and trade-offs.

**Questions:**

1. How does the proposed method support GQA?
2. Is the dynamic cache eviction compatible with FlashAttention? If so, how much decoding speedup does it achieve compared to standard FlashAttention?
3. Can the proposed approach be applied to enhance sparse attention methods such as Quest[2] or ShadowKV[3]?
4. In [1], the L1 norm is used to define perturbation theoretically, and experiments show little empirical difference between the L1 and L2 norms. It would be valuable to clarify the motivation for adopting the squared Frobenius norm in this paper and to discuss how it differs from the L1 norm in evaluating perturbations in practice.


Reference:
1. Identify Critical KV Cache in LLM Inference from an Output Perturbation Perspective
2. Quest: Query-aware sparsity for efficient long-context llm inference
3. Shadowkv: Kv cache in shadows for high-throughput long-context llm inference

---

> ### Author Response · Authors · 2025-11-25
> **Response (1/4)**
>
> Thank you for reviewing our paper and providing constructive feedbacks. In response to your comments, we address your concerns as outlined below:
>
> > W1: Theoretical presentation is poorly organized, with most details relegated to the appendix.
>
> Thank you for raising this concern. We have reorganized the theoretical contents in Section 3 of the revised manuscript to improve clarity, including introducing clearer definitions and theorem statements in the main text. The full proofs required to obtain the three OBCache scores involve detailed gradient and Hessian derivations, which are lengthy, so we have to keep them in Appendix B to preserve readability. We hope this improves the clarity of the theoretical contents.
>
> > W2: Discussion and comparison with relevant work.
>
> Thank you for pointing out the relevance of [1]. We apologize for not including it in our original submission. We were not aware of this work when conducting our study because it is very recent. We have now carefully reviewed it and appreciate the opportunity to clarify the differences.
>
> While both [1] and our work use output perturbation as a measure of token saliency, the core theoretical foundation and resulting methodology differ substantially. We want to clarify that output perturbation is a standard objective in layer-wise pruning, and within KV cache it has also been explored since CaM [2]. Our novelty is not in adopting this objective alone, but in formulating KV cache eviction as a structured pruning problem based on Optimal Brain Damage (OBD), which to the best of our knowledge has not been done previously. Furthermore, our theoretical analysis based on OBD therefore differs fundamentally from [1].
>
> Specifically, our analysis closely follows the OBD framework widely used in model pruning, which requires a Taylor expansion of the output perturbation with respect to the pruning units. Because the pruning units in KV cache are dynamic key and value matrices rather than static model parameters, deriving both first- and second-order information requires new theoretical effort, ultimately yielding three saliency scores (for key-, value-, and joint-pruning).
>
> In contrast, [1] derives an upper bound on output perturbation, leading to a closed-form expression depending only on $\mathbf{A}\mathbf{V}$. Consequently, [1] obtains a score equivalent to only one component of our results (the value-pruning score). Its method also requires a two-stage algorithm whose first step still selects tokens using attention weights, whereas our scores directly replace attention-weight heuristics under the structured pruning formulation. Moreover, [1] defines perturbation only on the current-step attention output, which does not extend to decoding-stage eviction, and does not consider the role of key matrices, which we show to be crucial.
>
> A key strength of our pruning-based formulation is its flexibility in choosing both the objective function and pruning variables. For example, our analysis does not assume a fixed sequence dimension for $\mathbf{O}$: it may represent a partial output (i.e., $\mathbf{O}_{w:s}$) or expand as decoding progresses. This enables both prefill- and decoding-stage pruning, naturally yielding the perturbation window that unifies differences among H2O, TOVA, and SnapKV (Section 3.4). Furthermore, while not explored in this work, our framework can be extended to unstructured or channel-wise pruning by redefining the pruning variables, or to cache merging by relaxing the diagonal Hessian assumption. Both are standard directions in model pruning literature. Therefore, our theoretical framework directly connects KV cache compression to model pruning and provides a principled foundation for future extensions.
>
> We hope this discussion clearly differentiates our work from [1] and and will clarify these distinctions in the revised manuscript.
>
> [1] Identify Critical KV Cache in LLM Inference from an Output Perturbation Perspective. Feng et al., 2025.
>
> [2] CaM: Cache Merging for Memory-efficient LLMs Inference. Zhang et al., 2024.

---

> > ### Author Response · Authors · 2025-11-25
> > **Response (2/4)**
> >
> > > W3: Evaluation of OBCache's computation overhead.
> >
> > We have added a detailed discussion in Appendix D.1 of the revised manuscript. We first provide a theoretical comparison of the runtime complexity of OBCache scores versus attention-based baselines:
> >
> > [W3-1]
> > | Method         | Complexity |
> > |--------------- |------------|
> > | Attn. Based    | $O(W)$|
> > | OBCache-Value  | $O(W + d_{\mathrm{hidden}})$|
> > | OBCache-Key    | $O(W d_{\mathrm{hidden}})$|
> > | OBCache-Joint  | $O(W d_{\mathrm{hidden}})$|
> >
> > Here, $W$ denotes the perturbation window size and $d_{\mathrm{hidden}}$ represents the per-head hidden dimension. Computing the value-pruning score only adds a small linear term in $d_{\mathrm{hidden}}$, whereas key- and joint-pruning require computing the norm of difference between each value vector and its corresponding attention output within the window, resulting in multiplicative complexity. However, in practical settings, $W$ is very small (e.g., $W=16$ in prefill and $W=1$ in decoding), so the additional cost remains minor. Additionally, the computations within the window are independent and inherently parallelizable, enabling efficient batched GPU execution that further mitigates the multiplicative factor in practice.
> >
> > We further benchmark the actual runtime of OBCache during both static and dynamic cache eviction, comparing prefill and decoding latency to attention-based baselines across different context lengths on a single A100-80GB GPU with batch size 1 (batch size 4 is reported in the revised manscript).
> >
> > [W3-2] Decoding (ms/token)
> > | Method         | 32k | 64k | 96k | 128k |
> > |--------------- |-----|-----|-----|------|
> > | Full KV        |68.66|133.2|192.5|OOM  |
> > | Attn. Based    |55.39|63.04|62.05|62.79|
> > | OBCache-Value  |56.45|62.93|63.35|64.82|
> > | OBCache-Key    |65.25|71.68|72.67|72.28|
> > | OBCache-Joint  |70.53|76.77|77.19|76.43|
> >
> > In decoding, we let the model generate 512 steps with varying context sizes, dynamically updating scores and evicting cache tokens at every step. We find that value-pruning score introduces negligible overhead (< 2 ms/token) relative to attention-based methods, and key- and joint-pruning scores introduce < 15 ms/token across all context lengths. Although the latter two are more expensive, they remain minor compared to the full-KV baseline as the context increases.
> >
> > [W3-3] Prefill (s)
> > | Method         | 32k | 64k | 96k | 128k |
> > |--------------- |-----|-----|-----|-----|
> > | Full KV        |3.556|9.634|18.44|30.02|
> > | Attn. Based    |3.654|9.903|19.03|30.72|
> > | OBCache-Value  |3.670|9.967|19.04|30.83|
> > | OBCache-Key    |3.698|10.02|19.10|30.90|
> > | OBCache-Joint  |3.732|10.04|19.15|30.95|
> >
> > In prefill, we measure the average seconds required from the start of prefill until the first token is generated, including the cache eviction operation. In this case, all eviction methods show negligible additional latency compared to the full-KV baseline. These empirical results align with the complexity analysis and demonstrate that OBCache scores can be efficiently computed.
> >
> > > W4: Reported improvements on real-world benchmark tasks appear marginal.
> >
> > Thank you for the observation. We would first like to clarify that the primary contribution of this work is the structured-pruning framework that formulates KV cache eviction from a theoretically grounded perspective, generalizing prior attention-weight scoring methods. This framework directly connects dynamic KV cache compression to model pruning and enables promising extensions.
> >
> > The empirical evaluation is intended to validate that OBCache scores (derived from an attention-output-level objective) are more effective than purely attention-weight scores (derived from a reduced attention-matrix-level objective). This is already supported by consistent improvements on baseline methods (H2O and TOVA) that rely solely on attention weights, where OBCache yields over 10% accuracy gain on NIAH and ~0.5 perplexity reduction on PG19.
> >
> > While performance gains on LongBench appear smaller, the benchmark comprises 16 heterogeneous datasets with diverse task patterns. OBCache scores are particularly effective in retrieval- and QA-oriented tasks (e.g., Qasper), and less impactful in summarization tasks where long-range retrieval plays a smaller role. Rather than pursuing state-of-the-art performance, this work focuses on establishing a principled framework for KV cache eviction. We view adapting OBCache for task-specific optimization and integrating it with more advanced eviction mechanisms as promising future directions.

---

> > > ### Author Response · Authors · 2025-11-25
> > > **Response (3/4)**
> > >
> > > > W5: Insufficient analysis or comparison among value-, key-, and joint-pruning scores in terms of their effectiveness and trade-offs.
> > >
> > > The three scores estimate pruning-induced eviction error at different levels of approximation. The joint-pruning score (Equation 3) considers both the individual and interactive effects of pruning key and value vectors simultaneously, and is therefore theoretically the most accurate estimator. The value-only and key-only scores can be viewed as decomposed components that isolate the individual contributions. Conceptually, pruning values affects only the weighted sum of value vectors, while pruning keys changes the entire attention distribution. Therefore, key pruning generally induces larger output perturbations than value pruning and is more sensitive.
> > >
> > > Empirically, we observe a consistent trend. Although key- and joint-pruning scores are more expensive to compute than the value-pruning score, they yield larger performance gains, particularly in the passkey retrieval experiments, where both key- and joint-pruning outperform value pruning across all three baselines.
> > >
> > > We hope this explanation clarifies the effectiveness and trade-offs among the three pruning scores.
> > >
> > > > Q1: How does the proposed method support GQA?
> > >
> > > Thank you for pointing this out. To support GQA, our experiments follow the implementation used in SnapKV, which stores KV cache for all query heads and performs eviction across them. However, the OBCache framework does support GQA without requiring this setup, due to its flexible objective formulation.
> > >
> > > Specifically, given per-query-head OBCache scores, per-KV-head scores can be obtained by a summation across heads within each KV group. This corresponds to using an objective that sums the output perturbation across all query heads sharing the same KV head, enabling eviction directly at the group level. The derivation is included in Appendix B.5 of the revised manuscript.
> > >
> > > Additionally, we evaluate this GQA-aware score and include the results of passkey retrieval and five tasks from LongBench in Appendix D.2. We find that adopting GQA-aware scores can greatly benefit performance in retrieval-centric tasks (e.g., NIAH and Qasper). However, on other tasks such as summarization and few-shot learning, the GQA setting leads to inferior performance.
> > >
> > > > Q2: Is the dynamic cache eviction compatible with FlashAttention?
> > >
> > > Unfortunately, cache eviction methods that rely on attention weights, including ours and all prior work such as H2O, TOVA and SnapKV, are incompatible with FlashAttention, because FlashAttention does not materialize the intermediate attention matrix, which is required to compute eviction scores. This limitation applies to both static and dynamic eviction.
> > >
> > > In our implementation, we follow SnapKV's approach: we use FlashAttention to accelerate the prefill computation, and then recompute attention scores at the query positions within the perturbation window. In decoding stage no FlashAttention is adopted. The decoding speedup mainly comes from the reduced kv cache.
> > >
> > > > Q3: Can the proposed approach be applied to enhance sparse attention methods?
> > >
> > > Thank you for raising this interesting point. Yes, our framework can be extended to enhance sparse attention methods. In that setting, the goal is to identify important token positions and perform selective attention while retaining all KV states. The pruning-induced eviction error analyzed in our work becomes exactly the true eviction error, so the OBCache saliency scores remain theoretically valid and potentially more informative, since there is no estimation gap in the objective.
> > >
> > > The main challenge is efficiently estimating the full attention matrix without explicitly computing it, since the complete KV cache can be very large in practice. Quest [1] addresses this by computing upper-bound attention scores based on channel-wise minimum and maximum of key vectors, although its saliency estimation still relies solely on attention statistics. We believe that similar attention estimation techniques can be incorporated into our output-aware OBCache scores, providing the possibility to improve token selection in sparse attention methods.
> > >
> > > [1] Quest: Query-Aware Sparsity for Efficient Long-Context LLM Inference. Tang et al., 2024.

---

> > > > ### Author Response · Authors · 2025-11-25
> > > > **Response (4/4)**
> > > >
> > > > > Q4: Motivation of adopting L2 norms and comparison with L1 norms.
> > > >
> > > > The choice of the L2 norm is primarily motivated by theoretical considerations. Our structured pruning framework requires computing first- and second-order derivatives with respect to the pruning units. While both L1 and L2 norms allow closed-form expressions in the value-pruning case, in key- and joint-pruning, the use of an L1 norm prevents obtaining a closed-form analytical solution. This is because the gradient and Hessian must be evaluated at zero, where the L1 norm is non-differentiable. In contrast, the squared L2 norm (equivalently the squared Frobenius norm) has well-defined gradients and Hessians, enabling closed-form analytical expressions for the resulting saliency scores. For this reason, L2 is widely adopted in classical layer-wise pruning literature.
> > > >
> > > > In addition to theoretical convenience, we also observe a slight empirical advantage of L2 over L1. We conduct addition experiments on the passkey retrieval (8K) task using LLaMA-3.1-8B-Instruct. We found that value-pruning scores with L2 norm mostly outperforms the L1 variant across compression rates, as shown below:
> > > >
> > > > | Method / KV Budget (# of tokens) | 80  | 160 | 320 | 400 | Avg. |
> > > > |--------------- |-----|-----|-----|------|-----|
> > > > | H2O                  | 10.8 | 47.2 | 75.2 | 79.2 | 53.1 |
> > > > | + OBCache-Value (L2) | 21.6 | 60.4 | 82.4 | 85.2 | 62.4 |
> > > > | + OBCache-Value (L1) | 19.6 | 57.6 | 81.2 | 85.6 | 61.0 |
> > > > | TOVA                 | 9.2  | 26.4 | 62.4 | 68.8 | 41.7 |
> > > > | + OBCache-Value (L2) | 10.4 | 37.6 | 71.6 | 81.2 | 50.2 |
> > > > | + OBCache-Value (L1) | 9.2  | 34.0 | 71.6 | 79.6 | 48.6 |
> > > > | SnapKV               | 58.4 | 92.0 | 96.4 | 97.2 | 86.0 |
> > > > | + OBCache-Value (L2) | 63.6 | 92.4 | 96.8 | 97.2 | 87.5 |
> > > > | + OBCache-Value (L1) | 49.6 | 93.2 | 97.2 | 97.2 | 84.3 |

---

### Author Response · Authors · 2025-12-03
**General Response and Clarification of Key Concerns**

We recognize and appreciate the actions taken by the ICLR committee in light of the recent circumstances. To facilitate the evaluation process, we provide below our general response and clarification of the key concerns commonly raised by the reviewers. We hope this provides helpful context for the area chair and others following our work.

**1. Discussion of relevant work** (raised by reviewers `MqZV`,`P3QT`)

Both reviewers pointed out that our submission did not cite [1], which also estimates token saliency from an output perturbation perspective. However, we later discovered that [1] is a concurrent work also under submission to this year’s ICLR. After carefully reading it, we clarify the key distinctions between the two works below.

- **(a) Summary of Our Theoretical Contribution.**
  Our work proposes the first theoretical framework that formulates KV cache eviction as a layer-wise structured pruning problem based on the Optimal Brain Damage (OBD) theory [2]. This formulation directly **connects KV cache compression to the model-pruning literature** and enables us to derive closed-form saliency scores for cache eviction. Importantly, output perturbation is a standard objective in layer-wise pruning [3,4], and it has also been explored within KV cache since CaM [5]. Our novelty does not lie in adopting this objective, **but in the OBD-based analysis**, which requires new theoretical effort to derive the first- and second-order Taylor terms with respect to the dynamic key and value matrices.

- **(b) Limitations of the Analysis in [1].**
  Unlike our direct second-order analysis, [1] characterizes token saliency by analyzing **upper bounds** of the output-perturbation objective rather than the perturbation itself. Specifically, in Theorem 3.3 of [1], the authors derive an upper bound whose expression includes the term $\mathbf{A}\mathbf{V}$ (corresponding to our value-pruning score). To isolate this term as a standalone score, their derivation in Theorem 3.5 then **further relaxes the bound** and invokes a **non-standard ad hoc assumption**: Assumption 3.4 in [1] posits that the top 50% of tokens, ranked by attention weight, collectively contribute more than 50% of the total attention mass. This heuristic condition is not supported by established theory and weakens the theoretical justification of their analysis. By contrast, our derivation **does not rely on bounding or relaxation steps**. We analyze the output perturbation directly using the standard OBD framework and do not introduce any additional ad hoc assumptions. Moreover, [1] defines perturbation only on the current-step attention output, which **does not extend to decoding-stage eviction**, and it does not consider **the role of key states**, which we show to be crucial.

- **(c) Advantages and Extensibility of Our Framework.**
  Because our framework is grounded in classical model-pruning theory, it naturally provides a principled foundation for **extending to more KV cache compression tasks**. This flexibility comes from the freedom to choose both the objective and the pruning units. For example, our analysis does not assume a fixed sequence dimension for the output matrix. This **enables both prefill- and decoding-stage eviction** and introduces the notion of a perturbation window, which **unifies accumulation strategies used by prior attention-based baselines**. Beyond token eviction, the same framework can be extended to unstructured or channel-wise pruning by redefining the pruning variables, or to cache merging by relaxing the diagonal-Hessian assumption [6]. Both are standard directions in the model-pruning literature.

Therefore, our theoretical contribution differs fundamentally from the approach taken in [1] and provides a more theoretically grounded and extensible foundation for KV-cache compression.

[1] Identify Critical KV Cache in LLM Inference from an Output Perturbation Perspective

[2] Optimal Brain Damage

[3] Sparsegpt: Massive language models can be accurately pruned in one-shot

[4] A simple and effective pruning approach for large language models

[5] CaM: Cache Merging for Memory-efficient LLMs Inference

[6] Second Order Derivatives for Network Pruning: Optimal Brain Surgeon

---

> ### Author Response · Authors · 2025-12-03
> **General Response and Clarification of Key Concerns (continued)**
>
> **2. Computational overhead of OBCache scores** (raised by all four reviewers)
>
> We provide a detailed complexity analysis and empirical evaluation in both our response and the updated manuscript (Appendix D.1). The results show that the additional overhead introduced by the OBCache scores is negligible during both prefill and decoding.
>
> **3. Rationale hehind the three OBCache scores** (raised by reviewers `MqZV`,`UVpj`)
>
> In Section 3.2, we first distinguish between the *true eviction error* (an unobservable quantity for all eviction methods) and the *pruning-induced eviction error* (a measurable surrogate) and clarify their respective roles in quantifying token saliency. In Section 3.3, because exactly computing the *pruning-induced eviction error* is infeasible, we perform a second-order Taylor expansion, which yields three closed-form OBCache scores that estimate it at different levels of approximation (corresponding to the components of Equation 3).
>
> **4. Choice of the perturbation window** (raised by reviewers `UVpj`,`P3QT`)
>
> The *perturbation window* is not a newly introduced hyperparameter, but rather a unifying notion that connects and formalizes the accumulation strategies used in prior methods. As discussed in Section 3.4, prior attention-based eviction baselines (H2O, TOVA, SnapKV) accumulate attention scores across different query positions; under our pruning framework, this corresponds to accumulating the *pruning-induced eviction error* across those positions. Therefore, our framework generalizes prior approaches by further introducing output-aware signals, enabling more informed eviction decisions that go beyond raw attention statistics.
>
> We also analyze the effect of window size in Section 3.5. When the perturbation window is small, the accumulated score aligns well with the *true eviction error*. As the window grows, early tokens accumulate disproportionately high scores, causing the proxy objective to deviate from the true eviction error. In such cases, enforcing a recent-retention window, as in H2O, can improve performance. A similar analysis was also performed in SnapKV (where the concept is referred to as the *observation window*). It reports similar LongBench performance when the window size is set to 16, 32, or 64.

---

### Meta-Review · Area_Chair_hqXp · 2026-01-04

**Summary:**

The paper introduces OBCache, a framework designed to optimize KV cache memory usage during Long-Context LLM inference. Deviating from traditional heuristics that rely solely on accumulated attention weights, OBCache formulates the eviction process as a layer-wise structured pruning problem grounded in the Optimal Brain Damage theory. By applying a second-order Taylor expansion to minimize the perturbation of attention outputs, the authors derive closed-form saliency scores for Value, Key, and Joint pruning. The method is evaluated on benchmarks such as Needle-in-a-Haystack and  LongBench, claiming to generalize existing attention-based methods while providing more accurate token saliency estimation.

Strength:
- The adaptation of the classic OBD framework to the dynamic problem of KV cache eviction is mathematically elegant.
- The method shows consistent accuracy improvements over vanilla attention-based baselines.

Weaknesses:
-  A major flaw is the lack of discussion and comparison with recent, highly relevant eviction metrics. Reviewers raised valid concerns regarding missing comparisons with works that also analyze cache eviction from an output perturbation perspective (e.g., [1]) or utilize alternative eviction metrics (e.g., [2]). The absence of these comparisons makes it difficult to assess the true novelty and effectiveness of the proposed OBD-based scores.
- The paper relies on relatively old baselines and ignores current state-of-the-art optimization methods beyond SnapKV, such as head-wise ([3], [4]) and layer-wise ([5], [6]) eviction strategies. The authors fail to demonstrate either superiority over or compatibility with these advanced budget allocation techniques. Even when compared with SnapKV, the improvements on LongBench are modest, raising doubts about the method's competitiveness against more recent approaches. Given the rapid iteration of this field, a comprehensive comparison is necessary.
- Lacked a thorough comparison of computational overhead against other efficient methods.

[1] Identify Critical KV Cache in LLM Inference from an Output Perturbation Perspective, 2025.2

[2] A Simple and Effective L2 Norm-Based Strategy for KV Cache Compression, emnlp 2024

[3] Not All Heads Matter: A Head-Level KV Cache Compression Method with Integrated Retrieval and Reasoning, iclr 2025

[4] Ada-KV: Optimizing KV Cache Eviction by Adaptive Budget Allocation for Efficient LLM Inference, nips 2025

[5] CAKE: Cascading and Adaptive KV Cache Eviction with Layer Preferences, iclr 2025

[6] PyramidKV: Dynamic KV Cache Compression based on Pyramidal Information Funneling, 2024.6

**Reviewer Concerns:**

Addressed: During the rebuttal, the authors provided additional data regarding computational overhead (latency/throughput analysis), scaling to ultra-long contexts (128k tokens), and clarifications on the perturbation window. These responses may resolve concerns about the method's runtime efficiency and stability.

Outstanding: However, the critical issue of experimental sufficiency remains unresolved. The refusal to benchmark against the strongest relevant competitors, specifically those using improved metrics and adaptive budget strategies, undermines the paper's value. While the theoretical contribution is acknowledged, the lack of a comprehensive comparison in a fast-moving field prevents acceptance.

**Reviewer Scores:**

The critical issue regarding experimental sufficiency remains unresolved. Since the authors did not provide the necessary discussions and comparisons with recent methods during the rebuttal, it is unlikely that the reviewers would raise their scores to meet the acceptance threshold. The overall consensus remains negative due to these persisting empirical gaps.

---

### Decision · Program_Chairs · 2026-01-26

Reject